

# Microwave Radar/radiometer for Arctic Clouds MiRAC: First insights from the ACLOUD campaign

Mario Mech[1], Leif-Leonard Kliesch[1], Andreas Anhäuser[1], Thomas Rose[2], Pavlos Kollias[1,3], and Susanne Crewell[1]

[1]Institute for Geophysics and Meteorology, University of Cologne, Cologne, Germany
[2]Radiometer-Physics GmbH, Meckenheim, Germany
[3]School of Marine and Atmospheric Sciences, Stony Brook University, NY, USA

**Correspondence:** Dr. Mario Mech, Institute for Geophysics and Meteorology, University of Cologne, Pohligstr. 3, 50969 Cologne, Germany (mario.mech@uni-koeln.de)

**Abstract.** The Microwave Radar/radiometer for Arctic Clouds (MiRAC) is a novel instrument package developed to study the vertical structure and characteristics of clouds and precipitation onboard the Polar 5 research aircraft. MiRAC combines a frequency modulated continuous wave (FMCW) radar at 94 GHz including a 89 GHz passive channel (MiRAC-A) and an eight channel radiometer with frequencies between 175 and 340 GHz (MiRAC-P). The radar can be flexibly operated using different chirp sequences to provide measurements of the equivalent radar reflectivity with different vertical resolution down to 5 m. MiRAC is mounted for down-looking geometry on Polar 5 to enable the synergy with lidar and radiation measurements. To mitigate the influence of the strong surface backscatter the radar is mounted with an inclination of about 25° backward in a belly pod under the Polar 5 aircraft. Procedures for filtering ground return and range side-lobes have been developed. MiRAC-P frequencies are especially adopted for low humidity conditions typical for the Arctic to provide information on water vapor and hydrometeor content. MiRAC has been operated on 19 research flights during the ACLOUD campaign in the vicinity of Svalbard in May/June 2017 providing in total 48 hours of measurements from flight altitudes > 2300 m. The radar measurements have been carefully quality controlled and corrected for surface clutter, mounting of the instrument, and aircraft orientation to provide measurements on a unified, geo-referenced vertical grid allowing the combination with the other nadir pointing instruments. An intercomparison with CloudSat shows good agreement in terms of cloud top height of 1.5 km and radar reflectivity up to -5 dBz and demonstrates that MiRAC is able to fill the gap in observing low level clouds with its more than ten times higher vertical resolution down to about 150 m above the surface. This is especially important for the Arctic as about 45 % of the clouds during ACLOUD showed cloud tops below 1200 m, i.e., the blind zone of CloudSat.



# 1 Introduction

In the rapidly changing Arctic climate (e.g., Serreze et al., 2009; Graversen et al., 2008), the role of clouds and associated feedback remain unclear (Osborne et al.; Wendisch et al., 2017). In particular, understanding the effect of mixed-phase clouds whose persistence is controlled by a complex interaction of microphysical, radiative, and dynamic processes is still challenging

(Morrison et al., 2012). Information on their vertical structure and phase partitioning which control their radiative impact (Curry et al., 1996) is currently available from the few ground-based profiling sites in the Arctic, e.g., Barrow, Alaska (Shupe et al., 2015), Ny-Ålesund, Svalbard (Nomokonova et al., 2018). The use of synergistic lidar and cloud radar measurements are key for the study of these cloud systems. Passive microwave measurements further provide information on the vertically integrated liquid water path (LWP). The profiling sites provide important long-term statistics, however, they might be limited in their

representativity for the wider Arctic.

Polar-orbiting, passive satellite imagery provides coverage of the Arctic region, however, the retrieval of cloud properties is challenged by the surface properties and suffer from limited vertical information. Active space-borne measurements by lidar and radar, i.e., by the combination of Cloud-Aerosol Lidar and Infrared Pathfinder Satellite Observation (?, CALIPSO;) and CloudSat (Stephens et al., 2008) have been fundamental in better understanding the vertical structure of clouds around the

globe. However, the CloudSat Cloud Profiling Radar (CPR) provides limited information in the lowest 0.75 to 1.25 km due to the presence of strong surface echo (Maahn et al., 2014; Burns et al., 2016), while the CALIPSO lidar observations are often fully attenuated by the presence of supercooled liquid layers. Using CALIPSO and CloudSat measurements Mioche et al. (2015) identified the region around Svalbard to be particularly interesting to study mixed-phase clouds as these show here a higher frequency of occurrence (55 %) compared to the Arctic average (30[% in winter and early spring, 50 [%] May to

October).

Airborne platforms have the advantage of high spatial flexibility and accessibility of remote places comparable to satellite observations. While a number of airborne campaigns have been performed in the Arctic (Andronache, 2018; Wendisch et al., 2018) the use of radar/lidar system in these aircraft campaigns is rather limited. One notable exception was during the Polar Study using Aircraft, Remote Sensing, Surface Measurements and Models, of Climate, Chemistry, Aerosols, and Transport

(POLARCAT) campaign in spring 2008, Delanoë et al. (2012a) studied an Arctic nimbo stratus ice cloud using the French airborne radar–lidar instrument in detail.

During May/June 2017 the Arctic CLoud Observations Using airborne measurements during polar Day (ACLOUD; Wendisch et al., 2018; Knudsen et al., 2018) aircraft campaign was performed as part of the ArctiC Amplification: Climate Relevant Atmospheric and SurfaCe Processes, and Feedback Mechanisms project ((AC)[3]; Wendisch et al., 2017). The research aircraft

Polar 5 and 6 of the Alfred Wegener Institute (AWI) operating from Longyearbyen, Svalbard, deployed a remote sensing and in-situ microphysics instrument package, respectively. Polar 5 was equipped with the Airborne Mobile Aerosol Lidar for Arctic research (AMALi; Stachlewska et al., 2010) and spectral solar radiation measurement already operated during the VERtical Distribution of Ice in Arctic clouds (VERDI; Schäfer et al., 2015) campaign. During ACLOUD, the remote sensing package was complemented by the novel Microwave Radar/radiometer for Arctic Clouds (MiRAC). In contrast to most other millimeter



radars employed on research aircraft (e.g., Radar Aéroporté et Sol de Télédétection des propriétés nuAgeuses (RASTA; Delanoë et al., 2012b), High-performance Instrumented Airborne Platform for Environmental Research (HIAPER) Cloud Radar (HCR; Rauber et al., 2016), Wyoming Cloud Radar (WCR; Khanal and Wang, 2015), High Altitude and LOng range research aircraft Microwave Package (HAMP; Mech et al., 2014)), which use short microwave pulses for ranging, the radar of the

MiRAC package employs a Frequency Modulated Continuous Wave (FMCW) radar. Thus a lower peak power transmitter is used, however, carefully consideration on handling the surface return is required. Therefore, in the past airborne FMCW radar has been mounted in uplooking geometry (Fang et al., 2016).

The purpose of this study is two-fold. First, the MiRAC package which consists of a unique 94 GHz FMCW radar (MiRAC-A) and an eight channel passive microwave radiometer with channels between 170 and 340 GHz (MiRAC-P) is introduced.

The instrument specifications and integration into the Polar 5 aircraft in downward looking geometry are provided in Sect. 2. The methodology used to quality control and mapping the observations to a geo-referenced coordinate system is described in Sect. 3. Second, the performance of the MiRAC during its first deployment within ACLOUD will be demonstrated first via a comparison with CloudSat within a case study in Sect. 4 and a statistical analysis of the ACLOUD measurements in Sect. 5. Conclusions and outlook to further analysis and deployments of MiRAC is given in Sect. 6.

## 15   2   Instruments and aircraft installation

MiRAC is composed of an active (MIRAC-A) and passive (MiRAC-P) part. MiRAC-A is mounted between the wings of the research aircraft Polar 5 and MiRAC-P is mounted inside of the aircraft measuring through a sufficient large aperture. Since, the FMCW radar needs a different measuring angle, MiRAC-A is tilted by 25 ° backwards with respect to nadir, whereas MiRAC-P is nadir-looking. The following three sections will describe the instruments and aircraft installation in detail.

### 20   2.1   FMCW W-band radar

MiRAC-A is based on the novel single vertically polarized cloud radar RPG-FMCW-94-SP manufactured by RPG-Radiometer Physics GmbH which is described in detail by Küchler et al. (2017). It basically consists of a transmitter with adjustable power to protect the receiver from saturation, a Cassegrain two-antenna system for continuous signal transmission and reception, and a receiver containing both the radar receiver channel at 94 GHz and the passive broadband channel at 89 GHz. To guarantee

accurate measurements both channels are thermally stabilized within a few mK. The FMCW principle allows to achieve high sensitivity for short range resolutions down to 5 m with low transmitter power of about 1.5 W from solid state amplifiers. The radar is also equipped with a passive channel at 89 GHz using the same antenna as the radar. The radar has been calibrated to provide the equivalent radar reflectivity $Z_e$ with an uncertainty of 0.5 dBz. The uncertainty of the brightness temperature ($TB$) measured by the 89 GHz channel is $\pm 0.5$ K (Küchler et al., 2017).

The cloud radar has originally been developed for ground-based application. Here the passive channel is especially useful because liquid water strongly emits at 89 GHz and with the cosmic background temperature as a low and well known background signal the LWP can be derived from TB measurements. As explained in the next subsection the strong and highly



variable emissivity of the surface complicates LWP retrieval from the airborne perspective. However, it additionally provides information about the presence of sea ice exploited from satellite (Spreen et al., 2008).

For the installation on the Polar 5 aircraft, the radar's antenna aperture size had to be reduced from 500 mm down to 250 mm in order to accommodate the radar into the Polar 5 belly pod. This implies a sensitivity loss of 6 dB compared to the original

RPG-FMCW-94-SP design. The smaller antenna size implies a wider half power beam width (HPBW) of 0.85° (antenna gain approx. 47 dB). The quasi bi-static system's 90 % beam overlap (beam separation of 298 mm) is achieved in a distance of 75 m from the radar (compared to 280 m for the 500 mm aperture radar). Therefore, measured $Z_e$ profiles start in 100 m distance from the aircraft.

In the case of aircraft deployments, the radar's receiver can be easily run into saturation caused by strong ground reflections

when pointing nadir, due to the fact that a FMCW radar continuously emits and receives signal power. A pulsed radar overcomes this problem, because the strong ground reflection pulse does not affect the atmospheric reflection signals, which are received delayed in time relative to the ground pulse. Therefore, the antenna axis of a down looking FMCW radar deployed on an aircraft must be tilted against the nadir axis, so that the ground reflection becomes significantly attenuated. A comprehensive analysis of sufficient inclination viewing angles relative to nadir for FMCW radar observations is given in Li (2005). The Polar

5 radar has been tilted by 25° from nadir backwards, following the guidelines in Li et al. (2005).

For an FMCW radar, ranging is achieved by transmitting saw tooth chirps with continuously increasing transmission frequency over a given sampling time and frequency bandwidth. The time difference between transmission and reception of a given frequency provides the range resolution. If the radar signal is backscattered by a particle moving towards or away from the radar an additional frequency shift much smaller than one from ranging occurs due to the Doppler effect. The Doppler spec-

trum for each range gate yields from the radar processing involving two Fast Fourier Transformations (FFT). For an airborne radar the Doppler spectrum is difficult to interpret due to the Doppler effect induced by aircraft motion (Mech et al., 2014). Although we can apply de-aliasing techniques to unfold the Doppler velocity, here we make only use of the equivalent radar reflectivity factor $Z_e$ which can be determined from the integral over the Doppler power spectrum.

During ACLOUD two different chirp sequences per profile defining the vertical resolution and thus minimum detectable $Z_e$

($Z_{min}$) were used to account for the fact that the sensitivity of the radar receiver decreases with the distance squared. With a range resolution of 17.9 m over the first 500 m as used for the first research flight (Table 1) $Z_{min}$ decreases from -65 dBz at 100 m distance from aircraft to about -50 dBz in a distance of 600 m (Fig. 1). Using a second chirp sequence with a range resolution of 27 m for the rest of the profile improves $Z_{min}$ which then again degrades with the distance squared reaching roughly -45 dBz at the typical flight altitude of 3 km above ground (Fig. 1). Based on the good performance achieved for these

settings the chirp sequences were modified twice during ACLOUD to achieve even finer range resolution (Table 1).

Figure 1 illustrates exemplary the actually achieved $Z_{min}$ for three research flights with the different settings. Herein, $Z_{min}$ is calculated for each profile using individual power measurements. In addition to the classical behaviour discussed above two features emerge. First, the strong surface backscatter yields saturation effect in the receiver. Due to different flight altitudes the decrease is spread over different range gates. Second, in some cases backscatter of hydrometeors or the surface echoes are

strong enough to shift $Z_{min}$ over the full profile.





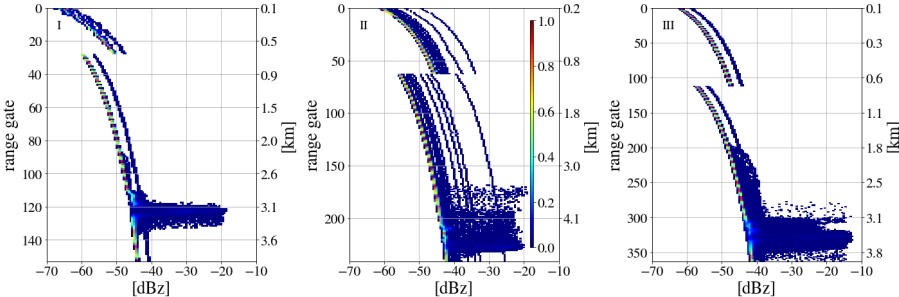

**Figure 1.** Sensitivity limit in [dBz] ($Z_{min}$) for vertical polarization of different chirp tables with different vertical resolution as a function of distance from the aircraft (secondary y-axes) for the three settings used during ACLOUD. The vertical resolution increases from a) to c) with increasing number of range gates, a) 154 range gates, May 25, 08:58 - 12:19 UTC, RF05, b) 242 range gates June 23, 12:53 - 13:43 UTC, RF22, c) 364 range gate May 27, 08:14 - 11:04 UTC, RF06.

## 2.2 Passive millimeter and sub-millimeter radiometer

In contrast to the MiRAC radar, the passive microwave channels deployed on the Polar 5 aircraft (RPG-LHUMPRO-243-340) are pointing nadir with respect to the aircraft fuselage. In order to co-align radar and passive observations, the atmospheric signal delay caused by the radar tilt must be taken into account by correcting for the aircraft's horizontal speed. For reference,

a detail description of MiRAC-P is provided below.

MiRAC-P consists of a double sideband (DSB) receiver with six channels centered around the 183.31 GHz water vapor (WV) line and two window channel receivers at 243 and 340 GHz. The schematic in Fig. 2 shows the overall system layout. The received radiation enters the radiometer through a low loss radome window (attenuation at 180 GHz approx. 0.01 dB) and is then reflected by an off-axis parabola antenna onto a wire grid beam splitter, forming beams between 1.3° and 1° (Table

2). The vertical polarization is transmitted into the 183.31 GHz water vapor receiver (WVR) while the horizontal polarization is further split in frequency by a dichroic plate, separating the 243 from the 340 GHz channel. All receivers are of DSB heterodyne type utilizing sub-harmonic mixers as the frontal element. The local oscillators (LOs) consist of Phase Locked Loop PLL stabilized fundamental dielectric resonant oscillators (DROs), multiplied by several active frequency multiplier stages as shown in Fig. 2. The frequency stability of these oscillators is close to $10^{-7}$ K$^{-1}$.

The WVR is equipped with a secondary standard, a noise switching system periodically injecting a precise amount of white noise power to the receiver input. By assuming a stable constant noise power over time, receiver gain fluctuations are effectively cancelled (noise adding radiometer, see Ulaby et al. (1981)). Unfortunately, state of the art noise sources with reasonable power output of at least 13 dB excess noise ratio are currently limited to maximum frequencies around 200 GHz, so that the two window channels (243 and 340 GHz) cannot use and benefit from them.

The WVR's Intermediate Frequency (IF) architecture is a six channel filter-bank design with the characteristics given in Table 2. All channels are acquired simultaneously (100 % duty cycle) by using a separate detector for each channel with 1 s temporal resolution. The window channel's IF bandwidth (BW) is 1950 MHz for both 243 and 340 GHz. Because of the DSB



mixer response, this corresponds to twice as much signal bandwidth of about 4 GHz (Table 2) having a small gap of 100 MHz in the center. Both sidebands are combined in the mixer IF output signal, so that a flat mixer sideband response is essential, meaning the mixer sensitivity and conversion loss must be almost identical in both sidebands. The subharmonic mixer design is optimal in this respect offering a sideband conversion loss balance of better than 0.1 dB. The most demanding receiver in

5   terms of sideband balance is the WVR due to its overall signal bandwidth of 15 GHz. The benefit of the DSB receiver design is a more than doubled radiometric sensitivity compared to a SSB (Single Sideband) receiver.

The parabolic mirror at the optical input can be turned to all directions for scanning purposes (sky view) or to point to the internal ambient temperature precision calibration target (accuracy 0.2 K). The WVR uses this target to determine drifts in receiver noise temperature while the 243 / 340 GHz channels are correcting for gain drifts. Typically, calibration cycles are

10   repeated automatically every 10 to 20 min by the radiometer's internal control PC. These long intervals are possible because of a dual stage thermal control system, stabilizing the receiver's physical temperatures to better than 30 mK over the whole environmental temperature range (-30 to +45°C). Given the receiver noise temperatures $T_R$ (Table 2) and the integration time of 1 s measurement noise is below 0.5 K.



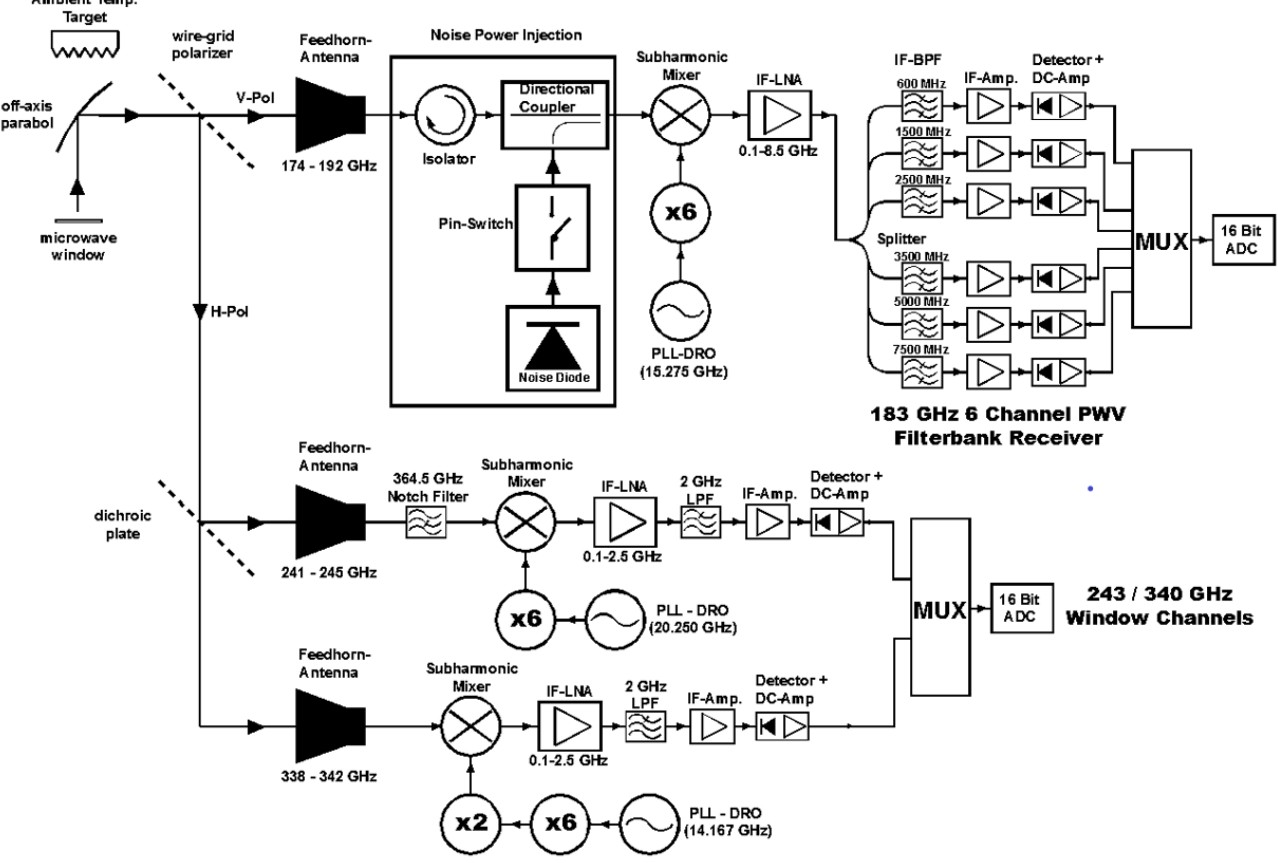

**Figure 2.** Block diagram of MiRAC-P.

## 2.3 Installation and Aircraft Operation

The Polar 5 aircraft is a Basler BT-67 operated by the Alfred Wegener Institute for Polar and Oceanic Research (Wesche et al., 2016). In addition to MiRAC, the AMALi lidar and radiation sensors were integrated into the Polar 5. To provide accurate information on the aircraft position an inertial navigation system is used which provides as well information on aircraft orientation, i.e., pitch $\epsilon$, roll $\rho$, and heading $\eta$ angles.

Due to the simpler electronic design and lack of high-voltage components compared to pulsed systems the FMCW radar has relatively small dimensions of $83 \text{cm} \times 57 \text{cm} \times 42 \text{cm}$ and weight of $88 \, \text{kg}$ allowing a relatively simple integration into the Polar 5 aircraft. As cabin space and openings are limited a special belly pod has been designed to accommodate MiRAC-A (Fig. 3) below the aircraft. The belly pod with a size of $200 \text{cm} \times 89 \text{cm} \times 50 \text{cm}$ has been designed and fabricated by Lake Central Air Services. Openings of 27 cm in diameter for transmitter and receiver antenna allows an unstopped view of MiRAC exposing the radomes directly to the environment. When grounded the aircraft fuselage is tilted by roughly $14°$ and the radar





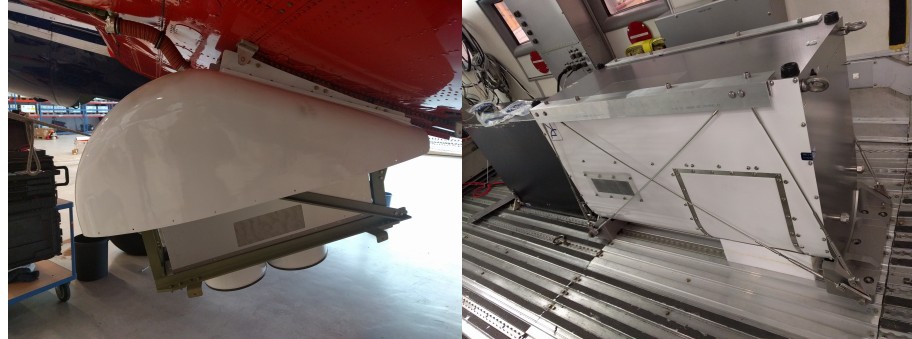

**Figure 3.** Left: MiRAC-A with opened belly pod below the research aircraft Polar 5. Right: MiRAC-P eight channel radiometer mounted in the aircraft cabin.

is integrated in the belly pod such that the pointing is about 25° backward during typical flight operation. The exact mounting position of the radar with respect to the aircraft is derived by a calibration method, which requires a calibration flight pattern, in which roll and pitch angle as well as flight altitude are changed rapidly over calm ocean. Further insight of determining the mounting position are described in Sect. 3.2.

In contrast to the MiRAC-A, MiRAC-P is integrated to Polar 5 roughly pointing at nadir during flight. While in ground based operation MiRAC-P can be mounted on a stand with the microwave transparent radome oriented towards zenith (Rose et al., 2005) here the radiometer box is fixed head over directly to the floor of the aircraft cabin (Fig. 3) looking through an opening in the fuselage. In this way the radome is directly exposed to the air avoiding any attenuation. In order to co-align radar and passive observations, the atmospheric signal delay caused by the radar tilt must be taken into account by correcting

for the aircraft's horizontal speed. To protect the instruments during start and landing the instrument compartment including MiRAC-P underneath the Polar 5 is protected via flexible roller doors.

For both passive components, MiRAC-P and the receiver at 89 GHz of MiRAC-A, absolute calibrations with liquid nitrogen have to be performed before the first flight after the installation as described in Rose et al. (2005) and Küchler et al. (2017). This procedure has to be repeated whenever the instruments are without power for longer period or are flown in significantly

different conditions. On ground the instruments are constantly heated to keep conditions stable for the receiver parts.

MiRAC has been operated successfully on 19 research flights (RF) with significant data loss occurring only during RF13 on June 5 2017 due to software problems. Though some flights were flown close to the ground for albedo and flux measurements, more than 50 % of the flight time was dedicated to straight legs above 2300 m altitude (pitch angle $\epsilon < 10°$ and roll $|\rho| < 3°$) allowing to probe a large range of different cloud conditions, e.g., over ocean, the marginal sea ice zone, and closed ice (Fig.

4). A special focus has been put on flights in the vicinity of the research vessel Polarstern that has set up an ice-floe camp North-West of Svalbard in the framework of the Physical feedback of Arctic boundary layer, Sea ice, Cloud and AerosoL (PASCAL) campaign (Wendisch et al., 2018) between June 5 and 14 2017.



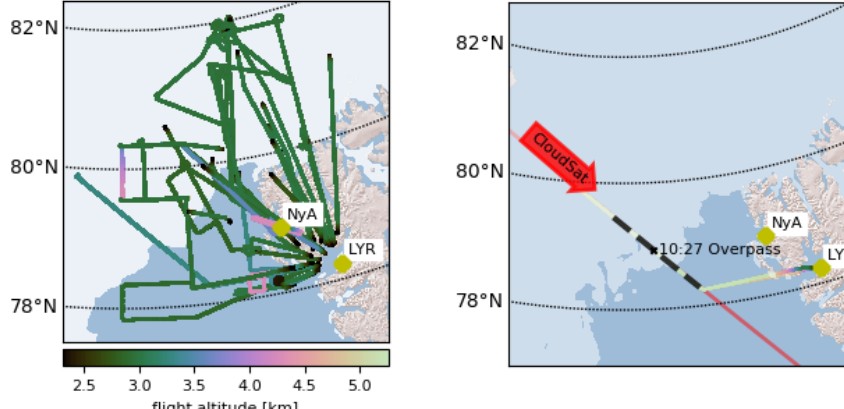

**Figure 4.** Left: tracks of all research flights of Polar 5 during ACLOUD around Svalbard with an altitude $h$ (above sea level) larger than 2300 km, $\epsilon < 10°$, and $|\rho| < 3°$. Right: Polar 5 CloudSat underflight on May 27 between 10:06 and 10:44 UTC West of Svalbard. In red the CloudSat track is shown. The white colored area shows the 15 % sea ice coverage derived from AMSR2 observations.

## 3 Data processing

All variables measured by MiRAC are recorded in the sensor-relative coordinate system. For scientific analysis, however, data with geographic coordinates longitude $\lambda$, latitude $\phi$ and altitude $h$ are needed. First, a methodology to identify and remove range side-lobe artifacts introduced by the strong surface echo return is developed (Sect. 3.1) and applied to the MiRAC-A observations on its native coordinate system. Lee et al. (1994b) provide an explicit analytical method to map data from aircraft-relative to local Earth-relative coordinates which we extended to fit our purpose (Sect. 3.2). In total five processing steps convert the raw data to the final geo-referenced data product (Table 3). Their effect for a radar time series is illustrated for a case study (Fig. 5).

### 3.1 Filtering of range side-lobes artifacts

The filtering described here identifies and removes non-meteorological artifacts in the radar reflectivity observations induced by range side-lobes. The slant distance of the aircraft to the surface can easily be identified from the range gate with the strongest $Z_e$ which is associated with the surface return. The strength of the surface radar return depends on the type of surface (land, sea-ice or sea) and wind speed. The FFT of piece-wise continuously differentiable functions lead to overshooting waves at discontinuities. This phenomenon is called Gibb's phenomenon (Gibbs, 1899; Gottlieb and Shu, 1997). In context to the strong surface radar reflectivity signal, range side-lobes can occur in range gates further away and lead to a contamination of the cloud profile. The effect depends on the filter characteristics of the FFT used in signal processing which typically produce symmetric side-lobes. While range gates above the surface can include contributions from both the atmosphere and the surface the "mirror signal" beyond the surface is only produced by the leakage of the surface return. This is illustrated for an one hour



time series in Fig. 5a. Clearly sub-surface reflection is visible in range gates beyond the surface especially in the first part of the flight with similar characteristics in the corresponding range gates above the surface. Note, that the second part of the flight leg is less affected which can be attributed to a change in surface characteristics in the marginal sea ice zone.

The first processing step I (Table 3) includes the removal of the mirror image , which is also called sub-surface reflection
filter. Herein the below surface range side-lobes are quantified and subsequently subtracted. For this subtraction of the mirror signal we assume that both range side-lobes above and below the surface are nearly similar which is justified by the symmetry of the digital FFT filter function. At each time step also the three measurements before and after are considered to locate the sub-surface reflection and its vertical extent. Within the located sub-surface reflection the value of the highest disturbance is used as subtracted value. The extent and distance from the highest signal of the surface to the center of the sub-surface reflection
provides the extent and distance to locate the range side-lobes above the surface. However, as illustrated in Fig. 5b still some scattered radar reflectivities remain. Thus, processing step II (Table 3) applies a speckle filter which removes isolated signals either remaining from the insufficient mirror image correction that does not take into account higher harmonics or are due to other processing artifacts. Most important thin isolated lines evident 5b need to be eliminated.

The speckle filtering is based on the procedure by Lee et al. (1994a). However, the filter is simplified by considering a radar
reflectivity mask, which is defined by setting all radar reflectivities to 1 and everything else to 0. Then, a box is defined to consider all neighboring measurements around a centered pixel. At a chosen threshold preferably close to 50 % of ones the centered value will be set to 0 or will be kept as 1. The aim of the filtering procedure is to remove single speckle pixel and horizontal disturbance lines. Thus, the box should be as small as possible and should have a rectangular shape tilted by 90° to the horizontal disturbance line comparable to the side-lobes. The value for the time-range is chosen as three because it is
the smallest value with a centered time step. Whereas the range-gate-range must be much larger than the time-range, but also an odd number. The observations show that the maximum extent of the disturbance line have an extent of five to six pixels in range-gate direction. This corresponds to a box with eleven or thirteen range gates, respectively, if the threshold for neighbors is close to 50 %. Taking thirteen range gates for the box gives a better opportunity to fit the threshold to the optimal exclusion of speckle and disturbance lines. Thus, empirical estimations lead to a threshold of 41.7 %. However, a data loss at cloud
boundaries always occurs by using such a filter. Figure 5c shows the result of the filtering procedure, which exclude speckle and horizontal disturbance lines.

Close to the surface the contamination by the surface reflection is too high to apply a correction. Therefore the lowest 150 m to the surface need to be ignored (Fig. 5f, grey shading). Further information of the filtered values can be found in the appendix (Table 6).





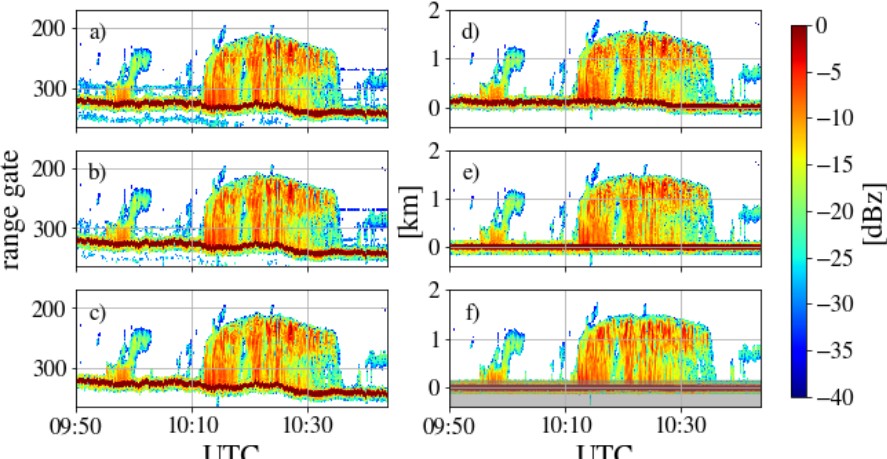

**Figure 5.** Time series of $Z_e$ profiles measured during RF06 on May 27 2017 for different processing steps (see Table 3): a) raw data, b) after subtraction of mirror signal, c) after speckle filter, d) filtered data on a time-height grid, e) corrected for sensor altitude, mounting position, pitch and roll angle, f) remapping onto a constant vertical grid. The grey shading indicates the range of surface contamination ($\leq 150\,\mathrm{m}$).

## 3.2 Coordinate transformation

For the conversion of the measurements into the geographical coordinate system the approach by Lee et al. (1994b) is extended and generalized. Two additional frames of reference are introduced. First, the sensor related coordinate system, in which the data are recorded and which is not identical to the platform (= airframe) coordinates. Second, the global geographic coordinate
($\lambda$, $\phi$, $h$) system, which is used in many applications and is of equal interest as the local Earth-relative coordinate (local East, North, zenith) system.

Then, the technique by establishing a mathematical object called *transform* that performs coordinate transformations between different reference frames is generalized. It can be inverted and composed, providing a simple formalism for multi-step coordinate transformations. Furthermore, it can be easily implemented in object oriented programming languages. The gener-
alization comes to the expense of a slightly elevated level of abstractness. A detailed description is provided in the appendix.

The coordinate transformation from the payload sensor-relative reference frame $X_s$ to the global geographic reference frame $X_g$, i.e., processing step III (Table 3), is done via two intermediate reference frames. First, the coordinates are transformed from $X_s$ to the platform-relative reference frame $X_p$. Then a transformation to the local Earth-relative reference frame $X_c$ is performed. Finally, the coordinates are transformed from $X_c$ to $X_g$. The origins and orientations of the reference frames are
defined in Table 4 and visualized in Fig. 6. If possible, the definitions of Lee et al. (1994b) are adopted.

The mathematical basis of the coordinate transformation and its application is described in detail in the appendix (A). Basically the mathematical operators $T_{ij}$ called *transforms* are defined which allow the simple conversion from one coordinate system into the next. In processing step IV (Table 3; Fig. 5d to e) the exact mounting of the sensor within the aircraft and the actual positioning of the aircraft are determined.



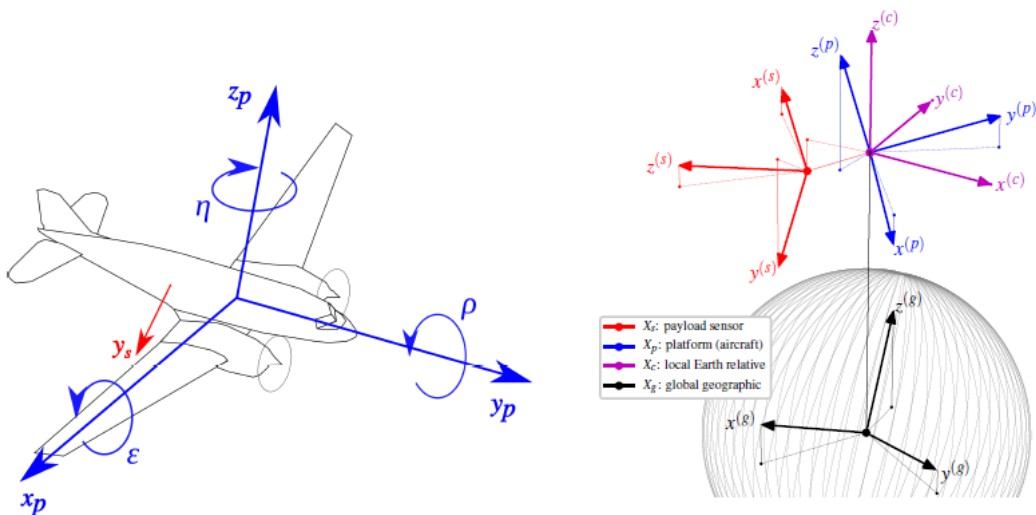

**Figure 6.** Left: sketch of the Polar 5 aircraft and the platform-relative $X_p$ reference frame. Right: reference frames for airborne measurements: sensor-relative $X_s$ (red), platform-relative $X_p$ (blue), local Earth-relative $X_c$ (purple), and global geographic (black). The grey lines are meridians of $X_g$ and the sphere they indicate may be seen as the planet surface, but distances are obviously not to scale. *Blue:* Coordinate axes of the aircraft reference frame $X_p$ and principal rotation angles: heading $\eta$, pitch $\epsilon$, and roll $\rho$. *Red:* $y$-axis of $X_s$.

The parameters that define $T_{sp}$, i.e., the transformation from the sensor to the platform reference frame, are the location and orientation of the payload sensor within $X_p$. Within the sensor installation (Sect. 2.3) these parameters were only known with moderate uncertainties ($\pm\,0.5$ m and $\pm\,3°$, respectively). Assuming that the position and attitude sensors of the Polar 5 operate on much higher precision, the other two transforms $T_{pc}$ and $T_{cg}$ are much more precise. The overall precision is thus limited

by $T_{sp}$. To get the precise sensor installation parameters, a calibration routine is developed. The calibration is performed over calm ocean or shallow sea ice in order to get a sharp discontinuity of the surface echo. Furthermore clouds shouldn't be too thick, so that the surface return of the radar is the strongest signal of the profile. The calibration assumes that the altitude of the signal maximum is the surface reflectivity return, which is at an altitude of 0 m. Due to variations in position and attitude of the platform, this is extremely unlikely to happen consistently when using wrong parameters.

A suitable time interval of 2.5 h over calm ocean surface is considered and the downhill-simplex algorithm of Nelder and Mead (1965) is applied. The algorithm is used to minimize the cost function $c = \sum_i \zeta_i^2$. This yields the position and attitude of the payload sensor relative to $X_p$. $\zeta_i$ is the altitude (in $X_g$) of the signal maximum at time step $i$ and $c$ is ideally equal to 0. However in practice, the minimum reachable value is bounded by the finite width of the sensor's range gates. Using this calibration, $c$ can be improved by a factor of three. Especially the attitude of the payload sensor has a large impact on

the transformed target altitude. Using the same technique, offsets in the interpretation of time readings between the payload,



position, and sensors attitude are detected in fractions of a second. These offsets affects $c$ because $T_{pc}$ and $T_{cg}$ are time dependent.

The performed calibration of the $z_s^{(p)}$ coordinate of the sensor position, the sensor attitude and the time offsets technique is stable with respect to changes of the first guess in a domain of reasonable estimates and the time interval chosen for the calibration. The parameters $x_s^{(p)}$ and $y_s^{(p)}$ show only very little effect on $c$. This is expected since most of the time they are close to orthogonal with respect to zenith. When including them in the calibration, the algorithm still converged in all investigated cases, but much slower.

Finally, the last processing step V (Table 3) shows the result of the remapping that interpolates the data onto a constant vertical grid. Herein the time shift of the tilted profile to a true vertical column is considered allowing an easy combination with the nadir pointing MiRAC-P, lidar and radiation data. The processed reflectivity data product is publicly available (Kliesch and Mech, 2019).

## 4 Case study

One of the objectives of the ACLOUD campaign is the evaluation of satellite products in the Arctic (Wendisch et al., 2018). Here, the added value of airborne radar observations is highlighted in this example of a CloudSat underflight that took place over the Arctic ocean northwest of Svalbard (Fig. 4). A roughly 30 min flight leg centered around the exact overpass time at 10:27 at 78.925°N and 2.641°E is shown in Fig. 7 together with the corresponding CloudSat measurements. Note that this stretch is also included in the processing example of Fig. 5 which allows a more detailed look into the MiRAC radar measurements which provides more than a factor of ten finer vertical resolution (< 30 m) compared to CloudSats 250 m data product. Note, that the resolution associated with the CloudSat pulse length is 485 m (Stephens et al., 2008). In terms of spatial resolution the 1.4 (1.8) km cross-track (along track) of CloudSat roughly corresponds to 30 MiRAC measurements (15 depending on aircraft speed).

The measurements are taken from a leg when the Polar 5 was flying south-east passing through the marginal sea ice zone towards the open ocean which is reached roughly at 78.6 °N as indicated by the sea ice product derived from the Advanced Microwave Scanning Radiometer (AMSR2) by the University of Bremen (Spreen et al., 2008). The transition from 100 % sea ice fraction in the beginning of the flight leg to open ocean at the end of the track is nicely seen by the change in the radar surface return which significantly increases in the vertical pointing CPR measurements close to the surface (Fig. 7). Note that here the surface contaminated range gates, i.e. blind zone, have not been eliminated. For MiRAC the lowest 150 m need to be omitted while for CloudSat the nominal blind zone is about 0.75 to 1.25 km depending on the surface echo strength (Tanelli et al., 2008). Nevertheless, the CPR detects the precipitating cloud system with maximum cloud top height of 1.6 km rather consistent in its spatial extent of (150 km) with MiRAC. In terms of reflectivity the CPR indicates slightly higher average values especially in the more southern part over ocean which however might result from additional surface contamination. Due to the low cloud top height we retain from looking at height averaged $Z_e$ profiles as done by (Delanoë et al., 2012a) for the case of a 5 km high nimbostratus cloud. As shown in Fig. 5 MiRAC is able to resolve the individual patches of enhanced





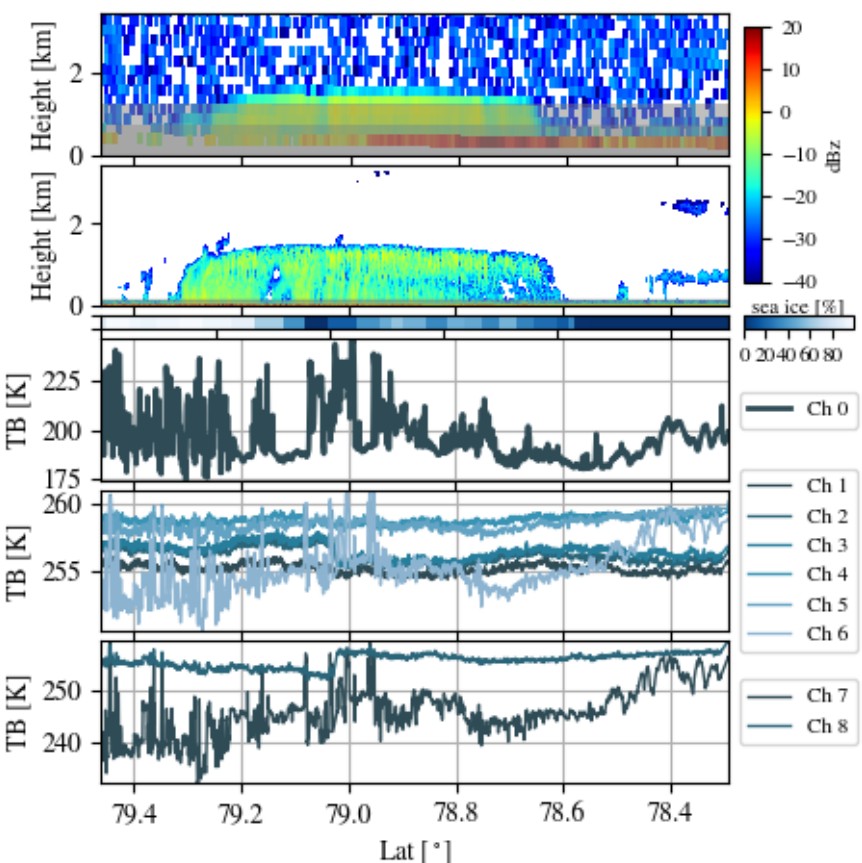

**Figure 7.** Vertical cross section of $Z_e$ measured by CloudSat CPR (top) and the MiRAC radar on Polar 5 (second row) for the satellite underflight on May 27 2017 between 10:06 and 10:44 UTC along the black dashed line in Fig. 4. Grey shaded areas define the zone of reduced sensitivity. The third row gives the sea ice coverage base on AMSR2 observations along the flight track. Row four to six show the passive radiometer measurements at 89 GHz (ch 0) from MiRAC-A and those channels of MiRAC-P, i.e., the six channels along the 183.31 GHz water vapor absorption line (Ch 1-6), and the two channels at 243 and 340 GHz (Ch 7 and 8).





reflectivities associated with turbulent processes as well as smaller scale clouds. Additional underflights were performed with CloudSat during ACLOUD unfortunately no CPR measurements are available due to satellite problems.

The daily sea ice product with 6.25 km spatial resolution mainly relies on TB measurements at 89 GHz. Such measurements are available with much finer resolution from MiRAC-A's 89 GHz channel. As can be seen in the beginning of the flight track

strong fluctuations in this channel between roughly 190 and 240 K mirror a strong change in surface emissivity (Fig. 7) with the lowest values being consistent with open water while higher TB indicate ice. These high frequency fluctuations are consistent with visual observations which reveal a high degree of brokenness in the sea ice. Towards the end TB stay at lower values typical for ocean surfaces before they increase again, however, with much smoother behaviour than during the broken sea ice conditions. This increase can be attributed to liquid water emission by the thin ($dz = 350$ m) cloud shown by the radar in

roughly 800 m height which can not be resolved by the CPR.

Time series of MiRAC-P TB clearly identify optically thick channels which are not affected by the surface by their relatively constant behaviour during the complete flight leg (Fig. 7). The first channel at 183±0.6 GHz being closest to the strong water vapor absorption shows the coldest TB as its emission stems from water vapor at higher altitudes. With channels moving farther away from the line center channels receive successively radiation from lower layers as the emission stems from lower

atmospheric layers. At a certain point along the line the atmosphere becomes transparent and surface emission also contributes to TB. This can be best seen for the outermost 183±7.5 GHz and the window channel at 243 GHz. This channel is of particular interest as it will also be flown on the Ice Cloud Imager (ICI; Kangas et al., 2014) onboard of MetOP-SG to be launched in 2023. Scattering by ice particles strongly increases with increasing frequency and therefore a brightness temperature depression can occur. Disentangling the contribution of water vapor, liquid water, the surface and ice scattering is complex and is part of

the ongoing retrieval development.

## 5   Cloud statistics

MiRAC as a remote sensing suite has been operated on Polar 5 during ACLOUD on 19 research flights which sums up to more than 80 flight hours. In a first analysis macroscopic cloud properties are derived for the full flight campaign. For that purpose the processed reflectivities (Sect. 3) taken in flight altitudes of at least 2300 m and with small aircraft orientation angles ($\epsilon < 10°$,

and $|\rho| < 3°$) are considered, resulting in usable 52 % of the total flight time along the tracks shown in Fig. 4. No measurements above land, i.e., the strong orography of Svalbard, are included. As seen in Fig. 8 about 80 % of all measurements considered in the statistical analysis have been acquired with flight altitude above 2800 m.

A radar cloud mask is defined by considering $Z_e$. A profile is attributed to be cloudy if a signal greater than $Z_{min}$ (Fig. 1) reaches a greater vertical extent than 25 m, which roughly correspond to two range gates in comparison to Table 1. The

cloud mask is then reduced to a one dimensional vector of ones and zeros describing clouds and clear sky, respectively. During the ACLOUD field campaign clouds occurred 75 % of the time (Table 5) which, however, has to be interpreted with care as flight patterns were selected to investigate clouds. Figure 8 provides the vertical cloud fraction resolved in 100 m intervals. It is highest in the lowest 1000 m with about 30 % and at heights above 2850 m. The latter results from cloud tops exceeding the





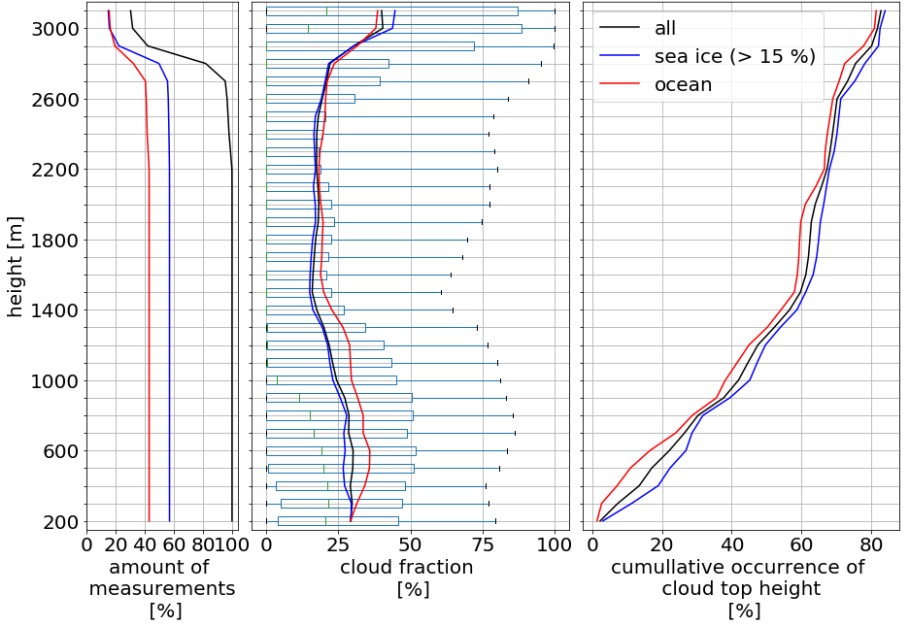

**Figure 8.** Height dependent cloud top height and cloud fraction (CF) on intervals of 100 m. The interval center is written in the y-ordinate: left: number of measurements , center: box-whisker plot of 20-min-CF with percentiles of 10, 25, 50, 75, and 90 % from left to right, respectively. The solid lines describe the total averaged cloud fraction, right: cumulative occurrence of cloud top height. The sea ice fraction is derived from satellite observations by AMSR2.

typical flight level of 10000 ft. In that cases the aircraft is forced to climb up, to get cloud profiles of the entire cloud. Figure 8 shows that about 30 % of the measurements were taken at altitudes higher than 3000 m.

In order to characterize the cloud variability within the grid cell of a global climate model, cloud fraction is calculated for 20 min legs. With a typical flight speed of 80 m/s this corresponds to roughly 100 km. The resulting distribution of cloud fraction for each height is shown in Fig. 8 in the form of box plots. Again highest variability with an interquartile range of 40 % or more occur in the lowest 500 to 1000 m above ground level associated with low clouds and above 3000 m due to the sampling. The radar signal is dominated by larger particles and therefore even few precipitating snow particles cause significant $Z_e$. Therefore the averaged cloud fraction in the lowest altitudes amounting to roughly 30 % is likely due to snow precipitation. Interestingly, below 500 m the spread in cloud fraction is decreasing towards the ground indicating the spatially rather constant occurrence of snow precipitation.

The radar cloud mask was used to derive information on cloud boundaries. This revealed that about 40 % of all clouds show cloud tops below 1000 m (Fig. 8) which are therefore likely to be missed by CloudSat. When looking at the vertical structure of clouds 62 (35) % appear to be single (two) layer clouds (Table 5) and even three or more layers are identified about three percent of the time. Looking at the thickness of these layer, not surprisingly, the multi-layer clouds show the shortest vertical extent (median $\Delta z = 205$ m) (Fig. 9). Most clouds have thicknesses below 1200 m which is consistent with the most frequent



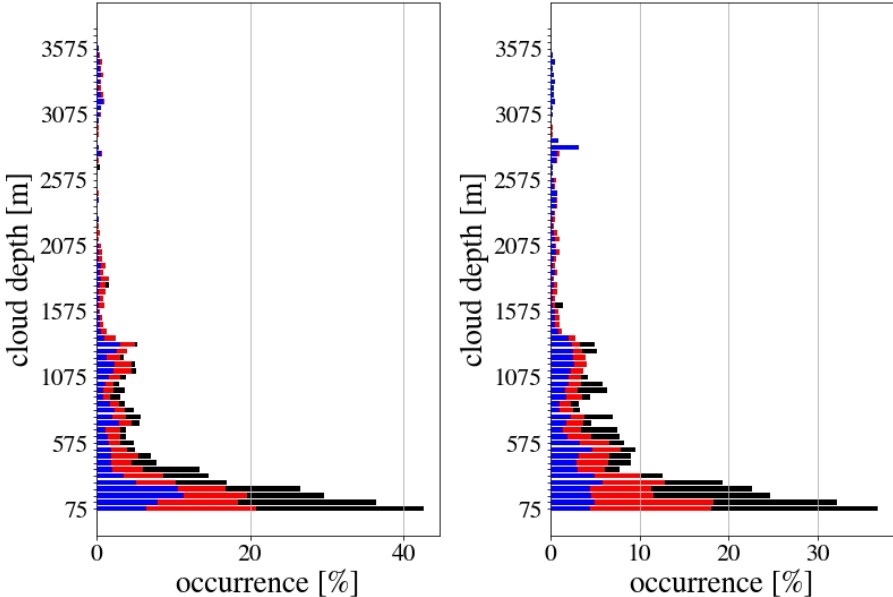

**Figure 9.** Cloud depth of different layered clouds. Blue describes the cloud depth distribution of all one layer clouds; red describes all cloud depths of two layer clouds, that is, the two layers results in two cloud depth values; black describes all cloud depths of clouds with more than three layers. The data are normalized such that all thickness bins of one type add up to 100 %. Left: sea ice surface (> 15 %), right: ocean surface.

cloud top heights (Fig. 8) and the frequent occurrence of precipitating clouds classical for arctic mixed-phase stratiform clouds. As discussed in Sect. 4 the information on liquid water from the passive channels can be used over open ocean to determine the LWP. In this way, together with AMALi and radiation measurements, detailed insights into mixed phase clouds will be gained.

In the beginning of the ACLOUD campaign a cold air outbreak could be observed which showed the classical behaviour of

a thickening boundary layer with higher cloud top heights when transitioning from the sea ice to the open ocean. During the Aerosol-Cloud Coupling And Climate Interactions in the Arctic (ACCACIA) campaign Young et al. (2016) investigated the microphysical structure of clouds during such a cold air outbreak and found largest number concentrations of liquid droplets over sea ice decreasing towards the ocean while ice characteristics did not change significantly. In a first statistical attempt all profiles observed during ACLOUD were separated into ocean and sea ice surface conditions using the sea ice concentration of

Bremen and a threshold of 15 %. The number of measurements above sea ice and broken sea ice is increased with respect to the number of measurements above ocean (Fig. 8). Figure 8 additionally shows less clouds above sea ice, which most frequently occur below 800 m. This is supported by Fig. 9, in which cloud thickness over sea ice is mostly lower than 800 m.

After analyzing the macrophysical properties, Constant Frequency by Altitude Diagrams (CFADs; Yuter and Houze, 1995) will now be considered, which provide the frequency of occurrence of $Z_e$ over the vertical profile. Figure 10 clearly shows the

much lower vertical extent of clouds over sea ice. The highest frequency for $Z_e$ occurs below 400 m between -20 and -10 dBz





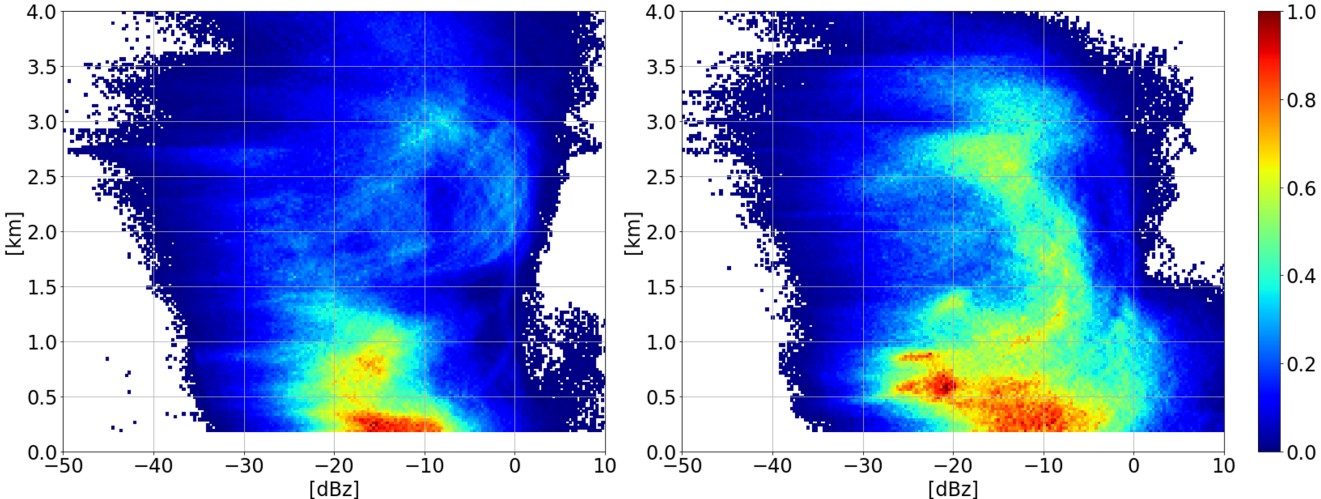

**Figure 10.** Contour Frequency by Altitude Diagrams CFADs of sea ice (left) and ocean (right). The frequency is normalized by the highest number within the CFADs for each case, respectively. A sea ice fraction of > 15 % is used (AMSR2).

indicating more frequent, but rather low amounts of precipitation. In contrast measurements over open ocean show a secondary peak of clouds clustering around -10 dBz in altitudes between 1 and 3 km. In general radar reflectivities are rather low with only few measurement over ocean showing higher reflectivities than 0 dBz emphasizing the need for a highly sensitive radar to observe Arctic low level clouds.

## 6   Conclusions and Outlook

The MiRAC is a novel airborne, active and passive microwave remote sensing instrument package with a 94 GHz FMCW radar and radiometers between 89 and 340 GHz. The instrument has been tailored to be fit into the Polar 5 aircraft and successfully participated in the ACLOUD campaign (Wendisch et al., 2018). A procedure to filter radar side-lobes and to provide geo-referenced data to the community has been developed. The preliminary data analysis from ACLOUD clearly demonstrates the capabilities of MiRAC especially for the study of low-level, mixed-phase Arctic clouds.

Deriving cloud microphysical properties from MiRAC and especially in synergy with other instruments, e.g., the AMALi lidar, operated on the Polar 5 will be the next step. As illustrated the passive channels are highly sensitive to sea ice allowing to determine the occurrence of sea ice with high spatial resolution. This, however, limits the possibility to retrieve cloud liquid water to open ocean. Exploring the information especially from the high frequency channels is of special interest in light of the upcoming MetOP-SG Ice Cloud Imager.

The Doppler spectra acquired by the MiRAC radar are difficult to interpret due to the influence of the aircraft motion on the Doppler shifts. Attempts to correct this are ongoing. Furthermore, information about multi-mode behaviour in the spectra can





also be used to better interpret the microphysics especially for those flights where in-situ measurements from the Polar 6 were performed.

During March/April 2019 MiRAC was part of the installation on Polar 5 in the Joint Aircraft campaign observing FLUXes of energy and momentum in the cloudy boundary layer over polar sea ice and ocean (AFLUX) flying out Longyearbyen on

Svalbard. In September 2019 the Multidisciplinary drifting Observatory for the Study of Arctic Climate (MOSAiC) campaign (http://www.mosaic-expedition.org) will start. MiRAC-P will be operated in up-looking geometry on board the research vessel Polarstern to infer moisture profiles in the central Arctic. In March/April and August/September 2020 flights with MiRAC-A and a downward looking Humidity And Temperature PROfiler (HATPRO; Rose et al., 2005) on board the Polar 5 aircraft will be performed again from Svalbard to further infer cloud characteristics in different seasons.

**Appendix A: Coordinate transformation**

First the mathematical basis of the coordinate transformation and the application to the experiment geometry followed by explicit coordinate transforms between the different reference frames discussed in Sect. 3 is provided.

**A1 Mathematical basis**

For the transition from one reference frame $X_i$ to another $X_j$ a mathematical operator $T_{ij}$ called *transform* is introduced. It

acts upon a position vector $\boldsymbol{r}^{(i)}$ in $X_i$-coordinates and returns its coordinates $\boldsymbol{r}^{(j)}$ in $X_j$:

$$\boldsymbol{r}^{(j)} = T_{ij}(\boldsymbol{r}^{(i)}). \tag{A1}$$

The vector is first rotated, then shifted:

$$T_{ij}(\boldsymbol{r}^{(i)}) = R_{ij} \cdot \boldsymbol{r}^{(i)} + \boldsymbol{S}_{ij}, \tag{A2}$$

where $R_{ij}$ is a matrix expressing the rotation of $X_i$ relative to $X_j$, $\boldsymbol{S}_{ij}$ is the position of $X_i$ in coordinates of $X_j$ and $\cdot$ is the

matrix product.

The inverse of the transform is obtained by solving Eq. (A2) for $\boldsymbol{r}^{(i)}$:

$$T_{ji}(\boldsymbol{r}^{(j)}) = R_{ij}^{-1} \cdot \boldsymbol{r}^{(j)} - R_{ij}^{-1} \cdot \boldsymbol{S}_{ij}, \tag{A3}$$

with $^{-1}$ being the matrix inversion operator (the inverse of a rotation matrix can be easily obtained by transposition). Equation (A3) has the same form as Eq. (A2), with rotation $R_{ji} = R_{ij}^{-1}$ and shift $\boldsymbol{S}_{ji} = \left(-R_{ij}^{-1} \cdot \boldsymbol{S}_{ij}\right)$.

The composition of two transforms $T_{ij}$ (from $X_i$ to $X_j$) and $T_{jk}$ (from $X_j$ further to $X_k$) yields the direct transform from $X_i$ to $X_k$. It is obtained by applying $T_{jk}$ to the result of $T_{ij}$:

$$T_{ik}(\boldsymbol{r}^{(i)}) = (R_{jk} \cdot R_{ij}) \cdot \boldsymbol{r}^{(i)} + (R_{jk} \cdot \boldsymbol{S}_{ij} + \boldsymbol{S}_{jk}), \tag{A4}$$

where $R_{ik} = (R_{jk} \cdot R_{ij})$ and $\boldsymbol{S}_{ik} = (R_{jk} \cdot \boldsymbol{S}_{ij} + \boldsymbol{S}_{jk})$ can be identified, respectively, as the rotation and shift of the composed transform.



## A2 Application to the experiment geometry

Once the three base transforms connecting the four reference frames $X_s$, $X_p$, $X_c$, and $X_g$ are established, the coordinates of the measurement targets can be transformed from $X_s$ to $X_g$ by applying the transform

$$T_{sg} = T_{cg} \circ T_{pc} \circ T_{sp} \tag{A5}$$

to the position vector $\boldsymbol{r}^{(s)}$ of the measurement (since measurements are only performed along the $y^{(s)}$-axis, $\boldsymbol{r}^{(s)} = r \cdot \boldsymbol{e}_y$) the transforms are obtained in principle.

The transform $T_{sp}$ from $X_s$ to $X_p$ is independent of time. It is described by the location of the payload sensor relative to the position sensor and by the orientation of the payload sensor relative to the sensor attitude. These relations are known from surveys before the campaign.

The time-dependent transform $T_{pc}$ from $X_p$ to $X_c$ is purely rotational as the two reference frames are co-located. It is described by the three principal rotation angles of the platform (Fig. 6). To reach $X_p$ from $X_c$, the coordinate system is first rotated by the (true) heading $\eta$ (distance to north) about the $z$-axis in mathematically negative sense. Then, a rotation about the $x$-axis by the pitch angle $\epsilon$ is applied (elevation of the nose). Finally, the system is rotated about the $y$-axis by the roll angle $\rho$. These three angles are recorded by the attitude sensor, which in practice is an inertial navigation system (INS) on board the aircraft.

The transform $T_{cg}$ from $X_c$ to $X_g$ $(\lambda, \phi, h)$ is time-dependent, too. It is done with knowledge of the platform position relative to the planet which is recorded by the position sensor (e.g., by use of a radio navigation-satellite service such as GPS). Since $X_c$ is aligned with the local east, north, and zenith, both shift and rotation of $T_{cg}$ are determined by the platform position.

## A3 From $X_s$ to $X_p$

The shift part of $T_{sp}$ is the sensor position in $X_s$ coordinates:

$$\boldsymbol{S}_{sp} = \boldsymbol{r}_s^{(p)}. \tag{A6}$$

As the $x^{(s)}$- and $z^{(s)}$-axes are undefined, the rotation is sufficiently described by two angles: The azimuth angle $\alpha_s^{(p)}$ measures how far the payload sensor's line of sight is rotated about the platform's $z^{(p)}$-axis away from the forward direction ($y^{(p)}$-axis); it is measured in mathematically negative sense (forward-right-backward-left). The view angle $\beta_s^{(p)}$ is the distance to 25    the negative $z^{(p)}$-axis (i.e., zero if looking downward w.r.t. the platform reference frame). The rotational part of $T_{sp}$ is the successive application of these two rotations:

$$R_{sp} = R_{sp,\alpha} \cdot R_{sp,\beta}, \tag{A7}$$

with

$$R_{sp,\alpha} = \begin{pmatrix} \cos\alpha_s^{(p)} & \sin\alpha_s^{(p)} & 0 \\ -\sin\alpha_s^{(p)} & \cos\alpha_s^{(p)} & 0 \\ 0 & 0 & 1 \end{pmatrix} \tag{A8}$$





$$R_{sp,\beta} = \begin{pmatrix} 1 & 0 & 0 \\ 0 & \sin\beta_s^{(p)} & \cos\beta_s^{(p)} \\ 0 & -\cos\beta_s^{(p)} & \sin\beta_s^{(p)} \end{pmatrix} \tag{A9}$$

## A4 From $X_p$ to $X_c$

Since the origins of the two reference frames are identical, this transform is purely rotational. The platform attitude relative to $X_c$ is described by the angles $\eta$, $\epsilon$, and $\rho$ (Sect. 3.2, the superscript $^{(c)}$ is omitted here). The transition from $X_p$ to $X_c$ is achieved by successively reversing these rotations:

$$R_{pc} = R_{pc,\eta} \cdot R_{pc,\epsilon} \cdot R_{pc,\rho}, \tag{A10}$$

with

$$R_{pc,\eta} = \begin{pmatrix} \cos\eta & \sin\eta & 0 \\ -\sin\eta & \cos\eta & 0 \\ 0 & 0 & 1 \end{pmatrix} \tag{A11}$$

$$R_{sp,\epsilon} = \begin{pmatrix} 1 & 0 & 0 \\ 0 & \cos\epsilon & -\sin\epsilon \\ 0 & \sin\epsilon & \cos\epsilon \end{pmatrix} \tag{A12}$$

$$R_{sp,\rho} = \begin{pmatrix} \cos\rho & 0 & \sin\rho \\ 0 & 1 & 0 \\ -\sin\rho & 0 & \cos\rho \end{pmatrix} \tag{A13}$$

## A5 From $X_c$ to $X_g$

Here, the platform position is used. It is usually recorded in the spherical coordinates longitude $\lambda_c$, latitude $\phi_c$, and altitude $h_c$ above mean sea level (superscript $^{(g)}$ is omitted here). Note that, because the origins of $X_c$ and $X_p$ coincide, $\lambda^{(c)} = \lambda^{(p)}$, $\phi^{(c)} = \phi^{(p)}$ and $h^{(c)} = h^{(p)}$. They sufficiently describe both the shift and the rotation of $T_{cg}$. The shift part of $T_{cg}$ is the platform position within $X_g$:

$$\boldsymbol{S}_{cg} = (x_c^{g)}, y_c^{(g)}, z_c^{(g)}), \tag{A14}$$





where $(x_c^{(g)}, y_c^{(g)}, z_c^{(g)})$ is the Cartesian representation of $(\lambda_c, \phi_c, h_c)$. The rotation matrix is established by first accounting for the latitude, then for the longitude:

$$R_{cg} = R_{cg,\lambda} \cdot R_{cg,\phi}, \tag{A15}$$

with

$$
5 \quad R_{cg,\lambda} = \begin{pmatrix} -\sin\lambda_c & \cos\lambda_c & 0 \\ \cos\lambda_c & \sin\lambda_c & 0 \\ 0 & 0 & 1 \end{pmatrix} \tag{A16}
$$

$$
R_{cg,\phi} = \begin{pmatrix} 1 & 0 & 0 \\ 0 & \sin\phi_c & -\cos\phi_c \\ 0 & \cos\phi_c & \sin\phi_c \end{pmatrix} \tag{A17}
$$

## 10   A6   From $X_s$ to $X_g$

A transform directly from $X_s$ to $X_g$ can be obtained by use of the composition formula in Eq. (A4):

$$T_{sg} = T_{cg} \circ T_{pc} \circ T_{sp}. \tag{A18}$$

This is conveniently done by a computing machine. The explicit form of $T_{sg}$ is not derived .

In order to obtain the target coordinates in spherical representation, the position vector in $X_g$ is eventually re-converted to
15  spherical coordinates after application of the transform.

*Data availability.* MiRAC-A radar reflectivity and brightness temperature data are available at PANGAEA database (https://doi.pangaea.de/10.1594/PANGAEA.899565).

*Author contributions.* SC conceptionalized MiRAC and initiated the DFG MiRAC and B03 project. TR designed and built MiRAC. MM organized all aspects of the aircraft integration and operation as well as the data analysis. PK advised on radar integration and processing. 20 AA developed the coordination transformation. LK developed the filtering procedure and conducted the statistical analysis. All co-authors contributed to writing the paper.

*Competing interests.* The authors declare that they have no conflict of interest.



*Acknowledgements.* MiRAC was acquried via grant INST 268/331-1 of the Deutsche Forschungsgemeinschaft (DFG, German Research Foundation). We gratefully acknowledge the funding by the DFG - project number 268020496 - TRR 172 "ArctiC Amplification: Climate Relevant Atmospheric and SurfaCe Processes, and Feedback Mechanisms (AC)³" in sub-project "B03: Characterization of Arctic mixed-phase clouds by airborne in-situ measurements and remote sensing". We thank the Alfred Wegener Institute for the support with the installation and operation of MiRAC on Polar 5. We thank Birte Kulla for her support preparing the manuscript.



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



**Table 1.** Chirp settings and corresponding range resolution for the different research flights (RF). MiRAC has been operated on 19 RF.

|  | I | II | III |
|---|---|---|---|
| Period | RF04, RF05 | RF19, 12:27 - 15:03 UTC | rest of RF |
|  |  | RF22, 12:53 - 13:47 UTC |  |
| percentage of occurrence [%] | 13 | 5 | 82 |
| range gate resolution first chirp [m] | 17.9 | 13.5 | 4.5 |
| number of range gates first chirp | 28 | 59 | 111 |
| extent of first chirp [m] | 500 | 800 | 500 |
| range gate resolution second chirp [m] | 27.0 | 22.4 | 13.5 |
| number of range gates second chirp | 126 | 183 | 253 |
| extent of second chirp [m] | 3400 | 4100 | 3400 |



**Table 2.** Specifications of MiRAC-P.

| channel | frequency [GHz] | bandwidth [MHz] | $T_R$ [K] | HPBW [deg] | gain [dB] |
|---------|-----------------|-----------------|-----------|------------|-----------|
| 1 | 183.31 ±0.6 | 200 | 1350 | 1.3 | 41.2 |
| 2 | 183.31 ±1.5 | 200 | 1350 | 1.3 | 41.2 |
| 3 | 183.31 ±2.5 | 200 | 1550 | 1.3 | 41.2 |
| 4 | 183.31 ±3.5 | 400 | 1300 | 1.3 | 41.2 |
| 5 | 183.31 ±5.0 | 600 | 1300 | 1.3 | 41.2 |
| 6 | 183.31 ±7.5 | 1000 | 1400 | 1.3 | 41.2 |
| 7 | 243 | 4000 | 900 | 1.25 | 43.6 |
| 8 | 340 | 4000 | 2100 | 1.0 | 45.4 |





**Table 3.** Processing steps for MiRAC-A radar measurements

| step | description | illustration in Fig. (5) |
|------|-------------|--------------------------|
| I | removal of mirror image | a) to b) |
| II | speckle filter | b) to c) |
| III | conversion from range to altitude system | c) to d) |
| IV | correction for sensor mounting and actual aircraft position | d) to e) |
| V | remapping onto constant vertical grid | e) to f) |





**Table 4.** Positions and orientations of the reference frames. The $x$- and $y$-axes of $X_g$ are defined in the common way: $x_g$ points towards the intersection of the equator and the prime meridian and the $y_g$ in the direction that completes the right-handed perpendicular coordinate system. Note that $X_c$ is *not* located on the planet's surface but on the platform.

| symbol | name | origin | $x$-axis | $y$-axis | $z$-axis | common coordinate name(s) |
|---|---|---|---|---|---|---|
| $X_s$ | sensor-relative | payload sensor | *arbitrary* | sensor direction | *arbitrary* | range |
| $X_p$ | platform-relative | platform | right wing | nose | stabilizer | right, forward, upward |
| $X_c$ | local Earth-relative | platform | east | north | zenith | east, north, zenith |
| $X_g$ | global geographic | Earth's center | *see caption* | *see caption* | North Pole | longitude, latitude, altitude |



**Table 5.** Properties of clouds detected above ocean, sea ice and both surface types.

|                                    | ocean | ice  | all  |
| ---------------------------------- | ----- | ---- | ---- |
| percentage of surface type[%]      | 56.5  | 43.5 | 100  |
| cloud fraction [%]                 | 80.1  | 72.0 | 75.5 |
| precipitation fraction [%]         | 36.0  | 37.9 | 37.1 |
| median CTH [m]                     | 1350  | 1260 | 1305 |
| mean CTH [m]                       | 1768  | 1683 | 1722 |
| percentage of 1 layer clouds [%]   | 65.3  | 60.0 | 62.4 |
| percentage of 2 layer clouds [%]   | 33.2  | 36.0 | 34.7 |
| percentage of $\geq$ 3 layers clouds [%] | 1.5   | 4.0  | 2.7  |



**Table 6.** Filter names and quality control of data in PANGAEA-files (Kliesch and Mech, 2019), description to variable "Ze flag", row 1 to 4 is already applied to get from "Ze unfiltered" to "Ze" and row 5 to 8 help for analyzing teh data.

| flag name | description |
|---|---|
| defective gate filter | increased reflectivity values in specific range gates are removed by a threshold |
| snr filter | anything below $Z_{min}$ is removed |
| speckle filter | side-lobe disturbances and speckle are removed |
| subsurface reflection filter | side-lobe disturbances are removed |
| quality disturbance possible range | possible range of side-lobes |
| quality surface influence range | range of surface contamination |
| quality disturbance in cloud | side-lobe disturbance in cloud (manually added) |
| quality disturbance | disturbance (manually added) |