# Peer review of "Microwave Radar/radiometer for Arctic Clouds MiRAC: First insights from the ACLOUD campaign"

_Atmospheric Measurement Techniques, 2019_

## Referee Comment (RC1) · Anonymous Referee #1 · 25 May 2019

This paper summarizes the performance of a new radar, a W-band airborne profiling FMCW radar. While I know how FMCW works, I have no actual practical experience with this kind of radar. My experience is with pulse-pair W-band profiling radars. Also, I am not going to comment on the high-frequency microwave radiometers to retrieve water vapor profiles in cold, dry environments, since that is not my expertise.

The main benefit of FMCW is that it can provide higher sensitivity and range resolution with a low power transmitter by utilizing it with close to 100% duty cycle. However FMCW systems can have issues that are difficult to avoid (see, for example, Delanoe et al, 2016 about their W-band FMCW radar BASTA), and clearly the authors are aware

of these issues.

Slanting the beam at about 25 deg from nadir, I expect, would indeed help with some reduction of the range lobes, but then you need to assume homogeneity in the horizontal in order to provide the vertical plane reflectivity. The paper explains why this slant angle approach is used, but it comes with a trade-off, essentially reducing both horizontal and vertical resolution. Given that the Polar 5 aircraft does not fly more than about 3 km AGL, this is perhaps not that big of a deal. On the other hand, based on the reported sensitivity (Fig. 1), the MIRAC should be able to probe clouds much farther than 3.5 km range, even though such data are not presented.

Personally I have more questions about radar performance issues (e.g., attenuation in liquid and in strong echoes, both of which are rare in the Arctic), but instead of showing more on the radar performance, the paper discusses the airborne radar data processing and multiple coordinate transformation at length. These things are rather standard and have been done before. Yes, I understand that given the slanted beam and wind installation, one need to know the 3D beam pointing angle quite correctly in order to end up with a vertical plane, but it is not as critical since this paper is not doing anything with the Doppler (at least in this paper). I am not sure why the authors discuss it at such great length instead of just mentioning the principle and some references, including the lengthy appendix for things that have been done and published before. Still, it is good to see the steps discussed systematically because it give confidence that it is done correctly, and then other papers can refer to this one.

This radar is an airborne version of a ground radar, and as far as I know, this is the first time a FMCW radar is deployed on an aircraft. For a ground based FMCW radar, a dual-antenna system may not be that much of an issue. For an airborne deployment, I think, bi-static can create problems not counting that you also need more space. With the increasing output power of the latest solid state amplifiers (SSA), I honestly do not see a reason to go with FMCW for airborne cloud radars. The paper cites 1.5 W for the MIRAC W-band SSA, today you can buy a 50 W W-band SSA. For FMCW

radars, the range lobes are a more serious issue than they are for a pulse radar using compression, but I am impressed that the ground contamination is limited to 150 m AGL, at least over the ocean.

This paper, while including some discussion on the radar and its sensitivity, is more descriptive about Arctic cloud properties from data collected during an Arctic campaign (and showing how much better this is than what CloudSat can see). That is fine for journal focusing on the science. But in a journal like Atmospheric Measurement Techniques, I expect to see more elaborating on the FMCW issues, especially for an airborne use, which is novel. For example, the paper describes the filtering they do, but presents just one figure/case to show that the filters generally do the job and get rid of the range lobes interference. Generalization is difficult to really evaluate: what about the areas where those lobes mix with strong weather echoes, and what about complex terrain? Thus there is not much to judge on the technical side of the radar even if I was experienced with FMCW technology and its issues. I do not have enough experience to judge the radiometer especially when it is combined with a radar, and I am wondering about possible interference between the 95 GHz radar and the 89 GHZ radiometer when used simultaneously.

In short, this paper nicely describes the MIRAC system, and it seems quite suitable to the thin, low reflectivity Arctic clouds and the low absolute humidity there. But the paper does not describe the radar performance in a broader context and for more diverse weather situations, and it mixes science (e.g. the layering of Arctic clouds) with instrument description. The latter should be the sole focus of papers in AMT.

---

## Author Comment (AC1) · 6 Jun 2019

We thank the reviewer for his time to carefully read the manuscript. However, we would like to clarify a few points.

It is important to state upfront that the integration of a 94-GHz FMCW radar on the Polar 5 is not the result of careful design for the aircraft selection (otherwise a higher altitude platform could have been selected) or the radar selection (where a single antenna, pulsed 94-GHz radar makes more sense). The research team was provided with one platform option and one available system that was designed for ground-based observations. Fortunately, the Polar 5 aircraft is large enough to accommodate a bistatic

radar system and we agree with the reviewer that a bistatic radar system is not generally preferred for aircraft deployment. The reviewer is certainly familiar with FMCW signal processing and appreciates the foresight of the research team to point the radar off-nadir by 25 degrees. As the reviewer points out, the surface echo side lobes are considerably reduced with this configuration.

Regarding the selection of an FMCW system, we would like to add that this particular radar system is not the first FMCW 94-GHz radar system. The Naval Postgraduate School (NPS) and Prosensing were operating a 94-GHz radar system on a twin otter for over a decade. One of the co-authors (Kollias) was involved in the analysis of the observations from this system and we agree that the surface echo affects significantly our ability to see weak targets. Solid state, high duty cycle transmitters are the future as it was clearly demonstrated with the launched of RainCube by JPL a smallsat that uses a pulse length of 160 microseconds and chirp to provide superior range resolution and excellent characterization of the transmitted waveform for optimum suppression of the surface return. In our case, we could not rely on such performance of the pulse compression and we employed post-processing that is explained in the manuscript.

One last reason why we do not describe the actual system in more detail is because the system is an exact copy of the ground-based version described in Kuechler et al. 2017.

Regarding the scientific content we like to note, that the manuscript is intended for the special issue of the ACLOUD campaign and provides important background for all using these data. Providing some simple macrophysical statistics is thus helpful in putting the measurements into context. Note that the prefered flight altitude of 10000ft in the unpressurized aircraft is due to limitation imposed by the lidar onboard, which is only certified up to this altitude. The issue of liquid water attenuation is important for the planned retrieval development which will also incorporate passive microwave and lidar measurements. Studies addressing this issue will be cited in a revision, e.g., Kuechler et al. 2018 or Meywerk et al. 2005.

On a moving plattform, the unfolding of the Doppler velocity is rather difficult. What would help is a sufficiently good background wind information, that could be used together with the aircraft speed. But unfortunately, no continuous wind profile measurements were available on board Polar 5, and the rather small number of dropsondes doesn't help with that. Using model data has been found to be to inaccurate. Any attempts that have been made so far were not very satifying. The only cases were we could derive somehow the Doppler velocity is for clouds that have been probed by the aircraft from different directions in rather short time periods.

We agree that over complex terrain the problem of filtering and removing the range lobes is more difficult. Especially the blind zone will be more investigated in future studies. Since we focus only on flights over open ocean, the marginal sea ice zone, and sea ice (all three areas shown in the case study Fig.7), this is not an issue for the measurements taken during the ACLOUD campaign, the recently performed ones during AFLUX, and the upcoming MOSAiC flights. So far, no case has been found in all flights where strong weather echos influenced our method of removing range lobes. We thank the author to point us to this issue and will take special care on this in future campaigns where we might experience stronger weather echos.

These arguments will be made more clear in a revised version of the paper.

**References**

N. Küchler, S. Kneifel, P. Kollias, and U. Löhnert. Revisiting Liquid Water Content Retrievals in Warm Stratified Clouds: The Modified Frisch. *Geophysical Research Letters*, 45(17):9323–9330, 2018.

N. Küchler, S. Kneifel, U. Löhnert, P. Kollias, H. Czekala, and T. Rose. A W-band radar-radiometer system for accurate and continuous monitoring of clouds and precipitation. *Journal of Atmospheric and Oceanic Technology*, 34(11):2375–2392, 2017.

J. Meywerk, M. Quante, and O. Sievers. Radar based remote sensing of cloud liquid water application of various techniques a case study. *Atmospheric Research*, 75(3):167–181, 2005.

---

## Referee Comment (RC2) · Anonymous Referee #2 · 7 Jun 2019

The study titled "Microwave Radar/radiometer for Arctic Clouds MiRAC: First insights-from the ACLOUD campaign" by Mario Mech et al. describes the deployment of a combined FMCW radar – microwave radiometer (MWR) platform onboard a research aircraft to study Arctic clouds.

This paper is divided into two parts: The first part (Section 2+3) is composed of the detailed description of aircraft-installation of the radar-MWR-instrument (named MiRAC) as well as the data processing to derive quality-controlled geo-referenced vertical profile observations. The second part (Section 4+5) focuses on measurements obtained during the ACLOUD field campaign conducted around Svalbard in May/June 2017 and

includes a case-study comparison with CloudSat observations.

The first part of Section 2 describes the FMCW radar system itself, the modifications of the ground-based version to the airborne system (basically, reduction of antenna size to fit into the aircraft at the expense of 6dB sensitivity and a wider half power beam width) and gives valuable information regarding issues arising during airborne downward-looking deployment of an FMCW radar and how they can be mitigated (off-nadir pointing by 25°) to reduce the ground echo influence. The FMCW radar principle is briefly illustrated and concludes with saying that this study focuses on the analysis of the equivalent radar reflectivity factor although "de-aliasing techniques to unfold Doppler velocity can be applied". - The reader is thus left wondering why this has not been done. (?) The capabilities of the FMCW radar allowing for different vertical range resolutions in different chirps are demonstrated for three different chirp programs, however only the characteristics of the first chirp program are discussed. It would be desirable to contrasts the pros and cons of all three used chirp programs.

The description of the passive MWR channels (MiRAC-P) is very technical and even includes a block diagram of the components. – Is this done in such a way because it is a first-time deployment of a novel instrument? If so, please state that clearly.

In Section 3 the different data processing steps are explained in a detailed way. In the radar signal, mirror images are removed and a speckle filter is applied. The description of the filter (p.10 lines 14-26) is sometimes a bit difficult to follow and could benefit from a re-read and some modifications to improve clarity. The multi-step coordinate transformation to convert from range to altitude is described in a straight-forward way and supported by the appendix. One quick question though: On p.12 line 3 it is mentioned that the sensor location is only known within +-0.5m. – This seems like a pretty large uncertainty. – What are the reasons for it?

In Section 4 a roughly 30min CloudSat overpass case study over different sea ice conditions is analyzed and the advantages of the lower blind zone and higher spatio-

temporal resolution of MiRAC is emphazised. The comparison also extends to comparing the brightness temperatures (TB) of the MiRAC-P to the AMSR2-TB-related sea ice concentration product highlighting the ability of MiRAC-P to detect small-scale features like broken sea ice which is not possible by the 6.25 km AMSR2 sea ice product resolution. This comparison is not mentioned in the abstract and should be added there.

Section 5 describes cloud statistics from 19 research flights during ACLOUD. This section can be improved by giving more reasons for surface-type (ice/open ocean) related differences in cloud altitude, observed number of cloud layers, cloud depth, and cloud reflectivities. The CFAD reflectivity plot (Fig 10) has another interesting feature: clouds over ocean exhibit a peak at 0.5-1km at very low reflectivities of below -20 dBz. – What's the explanation? Alternatively, Section 5 can be omitted since the paper has a good story line fitting AMT context which can finish after Section 4. Multiple previous ground-based remote-sensing based studies showing frequent occurrence of low-level Arctic clouds - as done in Section 5 - motivating the need of sensors being able to detect such low clouds already exist. I would suggest the manuscript to be published after minor revision addressing the above-and below points.

Minor comments

p.1 line 15: While it is important to fill the measurement gap of the CloudSat blind zone below 1.5km the phrase "MiRAC is able to fill the gap" seems a bit too strong since MiRAC is an aircraft-mounted instrument and thus limited in time and space and providing several tens of hours of observations during one field experiment instead of continuous coverage. . .please rephrase. p.2 line 3: Osborne et al. - publication year is missing p.2 line 6: Barrow is now called Utqiagvik p.2 line 7: add Summit, Greenland: https://esrl.noaa.gov/psd/arctic/observatories/summit/ p.2 line 13: missing citation p.2 line 22: Indicate how long the first airborne field experiments in the Arctic date back to p.2 line 25: ". . .Arctic nimbo stratus ice cloud observed during POLARCAT. . ." p.3 line 12: a "Second" without a "first" earlier on. . . p.3 lines 16-19: refer to the photograph/

of the placement of MiRAC-A and -P on the Polar 5 already here (Fig3) p.4 line 26: You mention the first chirp program is used for the first research flight – in Table 1 it is however stated that chirp setting "I" is used for RF04 and RF05. p.4 line 29: It sounds contradictory to state that based on the good performance of the chirp program "I" you modified it twice. . .why modify if performing well? p.4 lines 33-35: Be more precise how you identify the receiver saturation. The sentence "b. . .ackscatter of hydrometeors or the surface echoes are strong enough to shift Z_min over the full profile." is not clear – please clarify. p.8 line 16: add "during ACLOUD field experiment" p.9 line 18: replace "beyond" with "below" p.10 line 3: Second part of the flight is in the marginal sea ice zone. . .and the first part? p.11 line 11: "at" the expense p.15 line 13: add "AMSR2" before sea ice product p.15 line 29: 25m vertical resolution only refer to chirp program I in Table 1, correct? p.17: the "sea ice concentration of Bremen"? – There is sea ice in Bremen? ;) – Please correct. p.17 line 10: The sentence regarding "the number of measurements above sea ice "is increased" with respect to number of measurements above open ocean" is unclear. Do you mean "is higher"? p.17 line 12: Deriving a cloud depth over sea ice lower than 800m from Fig 9 seems a bit arbitrary. . .

Figures

Please check Figure quality (Fig6+7 have low resolution) and make sure all figures have proper axis labels with a variable and units (Fig1, Fig4, Fig5).

Fig1: Why is there an extra colorbar in the middle panel?

Fig 7: Add the Channel frequencies in the lower three panels to increase comparability between figure and description in the text.

[Figure]

---

## Author Comment (AC2) · 26 Jul 2019

We thank the reviewer very much for her/his detailed thoughts, the very useful comments, and suggestions on the manuscript, and thereby the possibility to further improve it. In the following we will address all major comments and list the changes we made in the manuscript. The section on the chirp tables and the cloud statistics have been remarkably modified to address the reviewers comments. The minor review points will be answered afterwards. In general, the manuscript has been revised and thereby strengthened according to the reviewers comments. Most of the figures have been modified according to the comments and now present the data in an easier

readable way. Text that has been revised or that has been added to the manuscript is written in italic letters.

**Major or general comments**

**The first part of Section 2 describes the FMCW radar system itself, the modifications of the ground-based version to the airborne system (basically, reduction of antenna size to fit into the aircraft at the expense of 6dB sensitivity and a wider half power beam width) and gives valuable information regarding issues arising during airborne downward-looking deployment of an FMCW radar and how they can be mitigated (offnadir pointing by 25°) to reduce the ground echo influence. The FMCW radar principle is briefly illustrated and concludes with saying that this study focuses on the analysis of the equivalent radar reflectivity factor although "de-aliasing techniques to unfold Doppler velocity can be applied". - The reader is thus left wondering why this has not been done. (?)**

We agree with the reviewer on this point, that the reader is left alone with the statement, because we only mention the possibility to apply de-aliasing techniques to unfold Doppler velocity without applying them and showing any results. As pointed out already in the answer to the first reviewer the unfolding of the Doppler velocity is rather difficult on a moving platform. Attempts to derive a somehow cleaned Doppler velocity gave unsatisfying results and it turned out, that a background wind information is necessary. Ideally, this comes from a continuous measurements like wind lidar. Using wind fields from models has been found to be to inaccurate. The only cases where we could derive somehow the Doppler velocity is for clouds that have been probed by the aircraft from different directions in rather short time periods. Other attempts to unfold the spectrum will be the subject of future studies and campaigns where we try to increase the number of drop sondes for the wind information.
To not leave the reader alone, as the reviewer said, we changed the sentence to:

*Although we can apply de-aliasing techniques to unfold the Doppler velocity, the results have not been satisfying so far. It has been found out, that background wind information is needed to disentangle the Doppler velocity from the aircraft motion. Such information is not available onboard Polar 5. Therefore, we make only use of the equivalent radar reflectivity factor Ze in this study which can be determined from the integral over the Doppler power spectrum.*.

**The capabilities of the FMCW radar allowing for different vertical range resolutions in different chirps are demonstrated for three different chirp programs, however only the characteristics of the first chirp program are discussed. It would be desirable to contrasts the pros and cons of all three used chirp programs.**

For the answers to reviewers comments on chirp table see our response to **p.4 lines 33-35**. There we describe the motivation of the three different chirp tables and include some describing text in the manuscript. This should address the question of pros and cons of the chirp tables and why we changed them during the campaign. In addition, the corresponding figure (Fig.1) has been modified so that it is now easier to compare the three chirp tables in terms of maximum range, range gates, and sensitivities.

**The description of the passive MWR channels (MiRAC-P) is very technical and even includes a block diagram of the components. – Is this done in such a way because it is a first-time deployment of a novel instrument? If so, please state that clearly.**

Indeed, it is the first description and deployment of a passive radiometer combining these frequencies in the millimeter and sub-millimeter range. Especially on an aircraft and in the Arctic this has never been done before. In the manuscript this point is now made clear in the beginning of the subsection by:

[Figure]

*The passive microwave radiometer MiRAC-P (or RPG-LHUMPRO-243-340) is a unique instrument combining millimeter and submillimeter channels that has been never operated before and especially not the in the Arctic and on an aircraft..*

**In Section 3 the different data processing steps are explained in a detailed way. In the radar signal, mirror images are removed and a speckle filter is applied. The description of the filter (p.10 lines 14-26) is sometimes a bit difficult to follow and could benefit from a re-read and some modifications to improve clarity.**

We agree with the reviewer. The paragraph describing the filtering was a bit difficult to read. We re-read the section and made some changes that should make it more easy to read an more understandable. The corresponding paragraph has been changed to:

*However, as illustrated in Fig. 5b still some scattered radar reflectivities remain. Thus, processing step II (Table 13) applies a speckle filter which removes isolated signals either remaining from the insufficient mirror image correction that does not take into account higher harmonics or that are due to other processing artifacts. Most important thin isolated horizontal disturbance lines evident in 5b need to be eliminated. The speckle filtering is based on the procedure by Lee et al. (1994a). However, the filter is simplified by considering a radar reflectivity mask, which is defined by setting all radar reflectivities to 1 and everything else to 0. Then, the filter uses a box considering all neighboring measurements around a centered pixel. At a chosen threshold preferably close to 50% of ones the centered value will be set to 0 or will be kept as 1. The aim of the filtering procedure is to remove single speckle pixel and horizontal disturbance lines, which may remain after processing step I. Thus, the box should be as small as possible and should have a rectangular shape tilted by 90° to the horizontal disturbance line comparable to the side-lobes. The value for the time-range is chosen as three because it is the smallest value with a centered time step.Whereas the range-gate-range must be much larger than the time-range, but also an odd number. The observations show that the maximum extent of the disturbance line have an extent of five to six pixels in range-gate direction. Having a filtering-threshold of 50% in mind,*

*the size of the box corresponds to eleven or thirteen range gates, respectively. Taking thirteen range gates for the box gives a better opportunity to fit the threshold to the optimal exclusion of speckle and horizontal disturbance lines. Thus,empirical estimations lead to a threshold of 41.7%. However, a data loss at cloud boundaries is obvious by using such a filter. Figure 5c shows the result of the filtering procedure, which exclude speckle and horizontal disturbance lines.*

**The multi-step coordinate transformation to convert from range to altitude is described in a straight-forward way and supported by the appendix. One quick question though: On p.12 line 3 it is mentioned that the sensor location is only known within +/-0.5m. – This seems like a pretty large uncertainty. – What are the reasons for it?**

The assumptions on the uncertainties given on p.12 line 3 are rather coarse since we do not know the exact position of the receiver in the radar/belly pod construction in relation to the aircraft center of mass. For the iterative process of finding the true position and alignment we made rather coarse assumptions to be sure it is within the range we search for the true values. In reality it is known with a higher accuracy. Therefore the statement has been changed to:

*Within the sensor installation (Sect. 2.3), these parameters are not exactly known and are therefore attributed with some uncertainties..*

**In Section 4 a roughly 30min CloudSat overpass case study over different sea ice conditions is analyzed and the advantages of the lower blind zone and higher spatio temporal resolution of MiRAC is emphasized. The comparison also extends to comparing the brightness temperatures (TB) of the MiRAC-P to the AMSR2-TB-related sea ice concentration product highlighting the ability of MiRAC-P to detect small-scale features like broken sea ice which is not possible by the 6.25 km AMSR2 sea ice product resolution. This comparison is not mentioned in the abstract and should be added there.**

We added to the end of the abstract a sentence for the sea ice observation with the MiRAC-A 89 GHz channel. The capabilities of the passive sensor of MiRAC-A for sea ice measurements are part of ongoing studies and will be explored in future campaigns.

*In addition, it is possible to get an estimate of the sea ice concentration by the 89 GHz passive channel of MiRAC-A with a much higher resolution than the daily AMSR2 sea ice product on a 6.25 km grid.*

**Section 5 describes cloud statistics from 19 research flights during ACLOUD. This section can be improved by giving more reasons for surface-type (ice/open ocean) related differences in cloud altitude, observed number of cloud layers, cloud depth, and cloud reflectivities. The CFAD reflectivity plot (Fig 10) has another interesting feature: clouds over ocean exhibit a peak at 0.5-1km at very low reflectivities of below -20dBz. – What's the explanation? Alternatively, Section 5 can be omitted since the paper has a good story line fitting AMT context which can finish after Section 4. Multiple previous ground-based remote-sensing based studies showing frequent occurrence of low-level Arctic clouds - as done in Section 5 - motivating the need of sensors being able to detect such low clouds already exist. I would suggest the manuscript to be published after minor revision addressing the above-and below points.**

We agree with the reviewer that omitting Section 5 and thereby reducing the manuscript to the instrument description and one measurement example would still be enough to be published in and fit the main focus of AMT. Nevertheless, we think that demonstrating the instruments and the setup capabilities by some simple statistical analysis on the observed Arctic clouds and precipitation over different surface types adds some valuable information to the manuscript. But again, as the reviewer says, it needs to be extended and some of the features shown in the figures need to be pointed out in more detail. Therefore, we revised Section 5 completely and included some more extended descriptions and explanations. More focus has been put on the differences between the surface types.

*... in Fig. 4 usable for the analysis. Due to the orography of Svalbard and the therefore difficult to interpret measurements, those above land are excluded. Most of the time Polar 5 was flying in an altitude of about 2900 to 3000m, which be seen as well in Fig. 8 where about 80% of all measurements considered in the statistical analysis have been acquired with this flight altitude or above. Figure 8 and Table 11 show as well, that about 57% of the measurements have been taken over open ocean and 43% over sea ice. It has to be kept in mind, that flight patterns have been planed to observe clouds according to numerical weather prediction models. Therefore, the statistics might be biased...*

*...the cloud fraction vertically resolved in 100m intervals. The highest values are present in the lowest 1000m with about 25 to 30% over sea ice and 30% and above over ocean (solid lines in Fig. 8). The cloud fraction is in general slightly higher over ocean than over sea ice in every height. For measurements at higher levels (above 2850m) the cloud fraction increases which is most likely an artefact since measurements at higher levels where only taken when Polar 5 was forced to climb above clouds due to cloud tops exceeding the typical flight level of 10000ft.*

Furthermore a more extensive description of the CFAD figures has been added. These shown interesting features that have been not mentioned so far.

*... low amounts of precipitation. A second cluster with lower amount can be found between 500 and 1000m with $Z_e$ values between -20 and -15dBz. Some higher reflectivities around 0dBz can be between 2 and 3km. In contrast measurements over open ocean show higher concentration of reflectivities in the lowest levels between -15 and -8dBz up to 500m and a secondary peak of clouds clustering -25 and -20dBz between 500 and 900m. This second peak not visible over is corresponds to the elevated arctic boundary layer height and the clouds forming here (Chechin and Lüpkes 2019). A band spanning from around -10dBz in 1km to -18dBz at 3km belongs to the vertical extending clouds over ocean. In general ...*

**Minor comments**

**p.1 line 15: While it is important to fill the measurement gap of the CloudSat blind zone below 1.5km the phrase "MiRAC is able to fill the gap" seems a bit too strong since MiRAC is an aircraft-mounted instrument and thus limited in time and space and providing several tens of hours of observations during one field experiment instead of continuous coverage. . .please rephrase.**

MiRAC measurements can be used to address the question how well CloudSat can sense the lower parts of the atmosphere, i.e., what is the amount of clouds and precipitation missed in low levels in comparison to MiRAC due to the larger blind zone, reduced sensitivity, and lower resolution. Therefore, filling the is more meant to be in terms of understanding and on a case study approach and not in terms of data coverage. We rephrased the sentence so it reads:

*...demonstrates that MiRAC with its more than ten times higher vertical resolution down to about 150m above the surface is able to show with some extend what is missed by CloudSat when observing low level clouds.*.

**p.2 line 3: Osborne et al. - publication year is missing**
year of publication has been added

**p.2 line 6: Barrow is now called Utqiagvik**
first paragraph changed to *...,e.g., Utqiaġvik (formerly known as Barrow), Alaska (Shupe et al., 2015), Ny-Ålesund, Svalbard (Nomokonova et al., 2018), Summit, Greenland (Shupe et al., 2013).*

**p.2 line 7: add Summit, Greenland:**
**https://esrl.noaa.gov/psd/arctic/observatories/summit/**
has been added (see above)

**p.2 line 13: missing citation**

added reference for CALIPSO *(CALIPSO; Winker et al., 2003)*

**p.2 line 22: Indicate how long the first airborne field experiments in the Arctic date back to**
We added: *While a number of airborne campaigns have been performed in the Arctic since the 1980's (Andronache et al., 2017; Wendisch et al., 2018)...*

**p.2 line 25: "...Arctic nimbo stratus ice cloud observed during POLARCAT..."**
This does not seem necessary since the sentence already mentions POLARCAT.

**p.3 line 12: a "Second" without a "first" earlier on**
has been changed

**p.3 lines 16-19: refer to the photograph of the placement of MiRAC-A and -P on the Polar 5 already here (Fig3)**
done

**p.4 line 26: You mention the first chirp program is used for the first research flight – in Table 1 it is however stated that chirp setting "I" is used for RF04 and RF05.**
sentence changed to: *... used for the first two research flights ...*

**p.4 line 29: It sounds contradictory to state that based on the good performance of the chirp program "I" you modified it twice. . .why modify if performing well?**
Indeed the motivation for the different settings was presented too briefly and has been revised – the new text is given below together with the next point of the reviewer.

**p.4 lines 33-35: Be more precise how you identify the receiver saturation. The sentence "b...ackscatter of hydrometeors or the surface echoes are strong enough to shift $Z_{min}$ over the full profile." is not clear – please clarify.**
The reviewer is right. We revised the whole paragraph to better explain our motivation for the different chirp settings and the characteristics of the receiver sensitivity. The new text reads as following:

*During ACLOUD two different chirp sequences per profile defining the vertical resolution and thus minimum detectable $Ze$ ($Z_{min}$) were used to account for the fact that the sensitivity of the radar receiver decreases with the distance squared. For the very first flights of MiRAC a conservative vertical resolution was chosen to ensure a high enough sensitivity even if unforeseen problems would arise. With a range resolution of 17.9m over the first 500m (Sequence I in Table 1) $Z_{min}$ decreases from -65dBz at 100m distance from aircraft to about -50dBz in a distance of 600m (Fig. 1). Using a second chirp sequence with a coarser range resolution of 27m for the rest of the profile improves $Z_{min}$ which then again degrades with the distance squared reaching roughly -45dBz at the surface for the typical flight altitude of 3km above ground (Fig. 1). Encouraged by the well-behaved performance of MiRAC with these conservative settings during the first flights the chirp sequences were modified to yield a higher vertical resolution of 4.5m in the first 500m and 13.5m for the rest of the flights (Sequence III in Table 1). Note, that due to higher flight altitudes the chirp settings had to be adapted (Sequence II) in Table 1) to still cover the full column during limited periods.*

*Figure 1 illustrates exemplary the actually achieved $Z_{min}$ for three research flights with the different chirp settings. Herein, $Z_{min}$ is calculated for each range gate by integrating over the noise power of the Doppler spectrum. Under typical atmospheric conditions this results in the classical behaviour discussed above. However, Figure 1 shows that sometimes deviations can occur which are due to the two following reasons: First, the Doppler spectrum noise power computation fails if the spectral width exceeds the range gate's maximum Nyquist velocity. This situation occurs in range gates affected by the strong surface reflectivity and causes the enhanced occurrence of $Z_{min}$ up to -20dBz. Due to different flight altitudes, e.g., clustered around 3.2km for the example in Fig. 1.a, enhanced $Z_{min}$ associated with the surface is spread over different range gates. Second the parallel shifts of $Z_{min}$ profiles are caused by the automatic transmitter power level switching. The radar automatically levels the transmitter power in cases when the input power might lead to receiver saturation effects. The signal power reduction when leads to reduced sensitivity over the whole profile leading to a over the*

*full profile. The automatic power reduction is triggered by high reflections which can occur under certain flight conditions, e.g., during flight maneuvers leading to a nadir viewing of the radar and thus increased surface backscatter.*

**p.8 line 16: add "during ACLOUD field experiment"**
done

**p.9 line 18: replace "beyond" with "below"**
done

**p.10 line 3: Second part of the flight is in the marginal sea ice zone...and the first part?**
The last two sentences of the paragraph have been changed to *Clearly sub-surface reflection is visible in range gates below the surface especially in the first part of the flight over sea ice (see Fig. 7 for sea ice cover) with similar characteristics in the corresponding range gates above the surface. The second part of the flight leg is less affected which can be attributed to a change in surface characteristics of the marginal sea ice zone and open water.* with including a reference to the figure showing the sea ice coverage of the flight section.

**p.11 line 11: "at" the expense**
changed

**p.15 line 13: add "AMSR2" before sea ice product**
done

**p.15 line 29: 25m vertical resolution only refer to chirp program I in Table 1, correct?**
It is true, this is only valid for chirp III. The sentence has been reformulated to: *...reaches a vertical extent greater than 25m, which roughly correspond to two range gates for chirp table III (or one range gate for chirp I and II, Table 11).*

**p.17: the "sea ice concentration of Bremen"? – There is sea ice in Bremen? ;) –**

**Please correct.**

That was misleading. Changed to *AMSR2* sea ice concentration.

**p.17 line 10: The sentence regarding "the number of measurements above sea ice "is increased" with respect to number of measurements above open ocean" is unclear. Do you mean "is higher"?**

Sentence changed to: *The number of measurements above sea ice and broken sea ice is higher than the number of measurements over open ocean (Fig.8).*

**p.17 line 12: Deriving a cloud depth over sea ice lower than 800m from Fig.9 seems a bit arbitrary...**

After reconsidering the statement we decided to remove it.

**Figures**

**Please check Figure quality (Fig6+7 have low resolution) and make sure all figures have proper axis labels with a variable and units (Fig.1, Fig.4, Fig.5).**

done

**Fig.1: Why is there an extra colorbar in the middle panel?**

It is not an extra colorbar. This colorbar gives the frequency of sensitivities. Unfortunately, most of the values are close to 0. Therefore, it looks like there is no connection between colorbar and data. We redesigned the figure. Now it is more clear.

**Fig.7: Add the Channel frequencies in the lower three panels to increase comparability between figure and description in the text.**

done with adapted caption. The channels numbers have been removed as well in Table 16, since they are obsolete now.